# SCORE-BASED DENSITY ESTIMATION FROM PAIRWISE COMPARISONS

**Petrus Mikkola**
Department of Computer Science
University of Helsinki
`petrus.mikkola@helsinki.fi`

**Luigi Acerbi**[*]
Department of Computer Science
University of Helsinki
`luigi.acerbi@helsinki.fi`

**Arto Klami**[*]
Department of Computer Science
University of Helsinki
`arto.klami@helsinki.fi`

## ABSTRACT

We study density estimation from pairwise comparisons, motivated by expert knowledge elicitation and learning from human feedback. We relate the unobserved target density to a tempered winner density (marginal density of preferred choices), learning the winner's score via score-matching. This allows estimating the target by 'de-tempering' the estimated winner density's score. We prove that the score vectors of the belief and the winner density are collinear, linked by a position-dependent tempering field. We give analytical formulas for this field and propose an estimator for it under the Bradley–Terry model. Using a diffusion model trained on tempered samples generated via score-scaled annealed Langevin dynamics, we can learn complex multivariate belief densities of simulated experts, from only hundreds to thousands of pairwise comparisons.

## 1 INTRODUCTION

Several complementary techniques, from flows (Rezende & Mohamed, 2015; Lipman et al., 2023) to diffusion models (Ho et al., 2020), can today efficiently learn complex densities $p(\mathbf{x})$ from examples $\mathbf{x} \sim p(\mathbf{x})$. With sufficiently large data, we can learn accurate densities even over high-dimensional spaces, such as natural images (Rombach et al., 2022). While challenges persist in the most complex cases, these models have achieved a high level of performance, proving sufficient for many tasks.

We consider the fundamentally more challenging problem of learning the density not from direct observations but solely from *comparisons of two candidates*. Given $\mathbf{x}$ and $\mathbf{x}'$ that are *not* sampled from the target $p(\mathbf{x})$ but rather from a distinct sampling distribution $\lambda(\mathbf{x})$ satisfying suitable regularity conditions, the task is to learn $p(\mathbf{x})$ from triplets $(\mathbf{x}, \mathbf{x}', \mathbf{x} \succ \mathbf{x}')$. The last entry indicates which alternative has higher density (the *winner* point). Being able to do this enables cognitively easy elicitation of subjective beliefs of an individual over random vectors. The canonical use-case is encoding expert knowledge into statistical models as prior information, with established literature in statistics dedicated to this problem of *prior elicitation* (O'Hagan, 2019; Mikkola et al., 2023). Here the belief is typically over a relatively low-dimensional space, but it needs to be inferred from a very limited number of observations to keep the expert effort manageable. Recently, elicitation tools have been increasingly used to quantify large language model (LLM) knowledge in probabilistic terms (Capstick et al., 2025; Requeima et al., 2024), for instance, to evaluate calibration or to use them as probabilistic cognitive models (Binz & Schulz, 2024) or forecasting models (Halawi et al., 2024). The current methods require dedicated techniques for direct prompting of probabilities, samples (Requeima et al., 2024) or moments (Capstick et al., 2025), whereas our formulation only requires comparative queries that LLMs can readily answer. Finally, the problem setup is also related to learning from human feedback (Ouyang et al., 2022), in particular to learning an implicit preference distribution for a generative model (Dumoulin et al., 2024).

---

[*]Equal contribution.

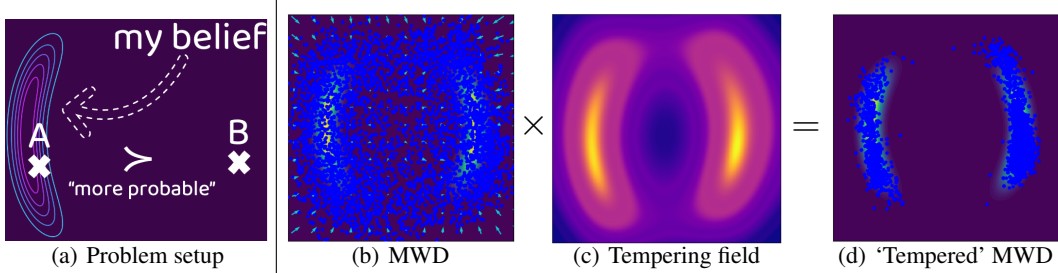

(a) Problem setup | (b) MWD | (c) Tempering field | (d) 'Tempered' MWD

Figure 1: (a) Problem setup. An expert holds a subjective belief over a parameter space, such as the likely hyperparameters of a learning algorithm (e.g. *learning rate* and *weight decay*), and can answer questions like *"Do you expect configuration A or B to work better?"*. We learn their belief as a density, to be used e.g. as a prior distribution for finding optimal hyperparameters. (b)-(d) Density estimation from 200 uniformly sampled pairwise comparisons, with the target density shown as a heatmap. (b) Samples and the score field at an intermediate noise level $\sigma$, for a diffusion model trained on the (winner, loser) pairs to model the marginal winner density (MWD). (c) Estimated tempering field. (d) Samples from the score-scaled annealed Langevin dynamics with the MWD score and a tempering field estimate. Samples align well with the target density, demonstrating the fundamental relationship between the scores of the estimable MWD and the latent target (belief density).

Recently, Mikkola et al. (2024) proposed the first solution for this problem, learning normalizing flows from pairwise comparisons and rankings. We propose an improved solution that also uses random utility models (RUMs; Train, 2009) for modeling the preferential data and is inspired by their idea of relating the target density $p(\mathbf{x})$ to a *tempered* version of the distribution of winner points, $p_w^\tau(\mathbf{x})$, for some *tempering parameter* $\tau \geq 1$. Since we have samples from $p_w(\mathbf{x})$, this relationship leads to practical algorithms once $\tau$ is estimated. In contrast to their empirically motivated heuristic link, we characterize this connection in detail and provide an *exact* relationship between the *scores* of $p(\mathbf{x})$ and $p_w(\mathbf{x})$. Since the relationship holds for the scores, it is natural to also switch to solving the problem with score-based models (Song & Ermon, 2019; Song et al., 2021), instead of flows. This brings additional benefits, for instance in modeling multimodal targets, and we empirically demonstrate a substantial improvement in accuracy compared to Mikkola et al. (2024). While they could learn densities from a modest number of rankings, they needed additional regularization to avoid escaping probability mass (Nicoli et al., 2023). Moreover, their best accuracy required more informative multiple-item rankings. In contrast, we focus solely on pairwise comparisons, which are easier to answer and more reliable (Kendall & Babington Smith, 1940; Shah & Oppenheimer, 2008), and widely used in AI alignment (Ouyang et al., 2022; Wallace et al., 2024).

Denote by $p_{\mathbf{x} \succ \mathbf{x}'}(\mathbf{x}, \mathbf{x}')$ the joint density of the available data, encoding the preferred candidate in the order of the arguments. The *marginal winner density* (MWD), denoted by $p_w(\mathbf{x})$, is obtained as its marginal as $p_w(\mathbf{x}) = \int p_{\mathbf{x} \succ \mathbf{x}'}(\mathbf{x}, \mathbf{x}')d\mathbf{x}' \propto \int \mathbb{P}(\mathbf{x} \succ \mathbf{x}')\lambda(\mathbf{x})\lambda(\mathbf{x}')d\mathbf{x}'$ where $\lambda(\mathbf{x})$ is the sampling density of the (independent) candidates. Our main theoretical contribution is a novel, *exact* relationship between the target $p(\mathbf{x})$ and the MWD $p_w(\mathbf{x})$ in terms of their scores: up to a reparameterization of the space, we have $\nabla \log p(\mathbf{x}) = \tau(\mathbf{x})\nabla \log p_w(\mathbf{x})$. Critically, $\tau(\mathbf{x})$ is not constant but a position-dependent *tempering field*. This implies we can perfectly recover $p(\mathbf{x})$ from the estimable $p_w(\mathbf{x})$ with score-based methods if the tempering field is known. We prove this foundational relationship for the popular Bradley–Terry model (Bradley & Terry, 1952; Touvron et al., 2023) and an exponential noise RUM, providing explicit formulas for the tempering fields.

Our second contribution is a practical algorithm derived from our theoretical insights. First, we propose to model the preference relationships by estimating the score of the joint density $p_{\mathbf{x} \succ \mathbf{x}'}(\mathbf{x}, \mathbf{x}')$, then train a continuous-time diffusion model (Karras et al., 2022) to recover the MWD by marginalizing it. Building on the ideal tempering field under the Bradley–Terry model, we estimate the tempering field $\tau(\mathbf{x})$ by using the analytical formula with importance samples from the trained MWD model and a simple density ratio model trained on the pairwise comparison data. Finally, we sample from the belief density $p(\mathbf{x})$ by running score-scaled annealed Langevin dynamics (Song & Ermon, 2019) with the MWD score and $\tau(\mathbf{x})$. Fig. 1 illustrates our approach.

## 2 BACKGROUND

### 2.1 DENOISING SCORE MATCHING AND ANNEALED LANGEVIN DYNAMICS

The (Stein) score of a probability density function $p(\mathbf{x})$, denoted $\nabla_{\mathbf{x}} \log p(\mathbf{x})$, is a vector field pointing in the direction of maximum log-density increase. Score-based generative methods approximate this score. They typically start by defining a family of perturbed densities $p_\sigma(\mathbf{x})$ by convolving $p(\mathbf{x})$ with noise at varying levels $\sigma > 0$; for example, $p_\sigma(\mathbf{x}) = p(\mathbf{x}) * \mathcal{N}(\mathbf{x}; \mathbf{0}, \sigma^2 \mathbf{I})$, where $*$ denotes convolution. A neural network $\mathbf{s}_\theta(\mathbf{x}, \sigma)$ with parameters $\theta$ is then trained to model the score of these perturbed densities, $\nabla_{\mathbf{x}} \log p_\sigma(\mathbf{x})$. This score network $\mathbf{s}_\theta$ is commonly trained through *denoising score matching* (Vincent, 2011), by minimizing the objective:

$$\mathcal{L}(\theta) = \mathbb{E}_{\mathbf{x} \sim p(\mathbf{x})} \mathbb{E}_{\sigma \sim p_{\text{train}}(\sigma)} \mathbb{E}_{\tilde{\mathbf{x}} \sim p_\sigma(\tilde{\mathbf{x}}|\mathbf{x})} \ell(\sigma) \left\| \nabla_{\tilde{\mathbf{x}}} \log p_\sigma(\tilde{\mathbf{x}}|\mathbf{x}) - \mathbf{s}_\theta(\tilde{\mathbf{x}}, \sigma) \right\|^2. \quad (1)$$

Here, $\tilde{\mathbf{x}}$ is a noisy version of a clean sample $\mathbf{x}$, generated via the perturbation kernel $p_\sigma(\tilde{\mathbf{x}}|\mathbf{x})$ (e.g., an isotropic Gaussian $\mathcal{N}(\tilde{\mathbf{x}}; \mathbf{x}, \sigma^2 \mathbf{I})$). The network is trained to predict the score of $p_\sigma$ by minimizing the objective in Eq. 1, where the perturbation kernel is typically tractable. The function $\ell(\sigma)$ provides a positive weighting for different noise levels. The noise levels $\sigma$ are drawn from a distribution $p_{\text{train}}(\sigma)$ following either a discrete, often uniform schedule $(\sigma_t)_{t=1}^{T}$ (Song & Ermon, 2019), or a continuous one (Karras et al., 2022).

Once trained, $\mathbf{s}_\theta(\mathbf{x}, \sigma)$ enables sampling from an approximation of $p(\mathbf{x})$. One prominent method, besides reverse diffusion processes (discussed later), is *annealed Langevin dynamics* (ALD) (Song & Ermon, 2019). ALD starts with samples $\mathbf{x}_T^{(0)}$ from a broad prior (e.g., $\mathcal{N}(\mathbf{x} \mid \mathbf{0}, \sigma_{\max}^2 \mathbf{I})$) and iteratively refines them. It runs $L$ steps of Langevin MCMC per noise level $\sigma_t$ along a decreasing schedule $\sigma_{\max} = \sigma_T > \ldots > \sigma_1 = \sigma_{\min}$:

$$\mathbf{x}_t^{(l)} = \mathbf{x}_t^{(l-1)} + \epsilon_t \, \mathbf{s}_\theta(\mathbf{x}_t^{(l-1)}, \sigma_t) + \sqrt{2\epsilon_t} \, \mathbf{n}_t^{(l)}, \quad l = 1, 2, \ldots, L, \quad (2)$$

with step size $\epsilon_t > 0$ and $\mathbf{n}_t^{(l)} \sim \mathcal{N}(\mathbf{0}, \mathbf{I})$. For $t < T$, $\mathbf{x}_t^{(0)} = \mathbf{x}_{t+1}^{(L)}$. Under ideal conditions ($L \to \infty$, $\epsilon_t \to 0$, accurate $\mathbf{s}_\theta$), $\mathbf{x}_1^{(L)}$ approximates a sample from $p_{\sigma_{\min}}(\mathbf{x}) \approx p(\mathbf{x})$ (Welling & Teh, 2011).

### 2.2 DIFFUSION MODELS

A continuous-time diffusion model describes a forward process that gradually transforms a data distribution $p(\mathbf{x})$ into a simple, known prior distribution (e.g., a Gaussian). This process is often defined by a forward-time stochastic differential equation (SDE) (Song et al., 2021):

$$d\mathbf{x} = f(\mathbf{x}, t)dt + g(t)d\mathbf{b},$$

where $\mathbf{b}$ is Brownian motion, $f(\mathbf{x}, t)$ is the drift coefficient, and $g(t)$ is the diffusion coefficient. If $\mathbf{x}(0) \sim p(\mathbf{x})$ (the target density), its time-evolved density is $p_t(\mathbf{x})$. If $f$ is an affine transformation, then the transition kernel $p(\mathbf{x}(t)|\mathbf{x}(0))$ is Gaussian and for a sufficiently large $T > 0$, the marginal distribution $p_T(\mathbf{x}(T))$ becomes a pure Gaussian, such as $\mathcal{N}(\mathbf{0}, \mathbf{I})$ or $\mathcal{N}(\mathbf{0}, T^2 \mathbf{I})$.

The forward process can be reversed to generate data. Starting from a sample $\mathbf{x}_T \sim p_T(\mathbf{x})$, one can obtain a sample $\mathbf{x}_0 \sim p(\mathbf{x})$ by solving the corresponding reverse-time SDE (Anderson, 1982):

$$d\mathbf{x} = \left( f(\mathbf{x}, t) - g^2(t) \nabla_{\mathbf{x}} \log p_t(\mathbf{x}) \right) dt + g(t)d\bar{\mathbf{b}}, \quad (3)$$

where $\bar{\mathbf{b}}$ is Brownian motion with time flowing backward from $T$ to 0. Alternatively, samples can be generated by solving the deterministic probability flow ODE (Song et al., 2021),

$$d\mathbf{x} = \left( f(\mathbf{x}, t) - \frac{1}{2} g^2(t) \nabla_{\mathbf{x}} \log p_t(\mathbf{x}) \right) dt. \quad (4)$$

Both reverse methods require the score function $\nabla_{\mathbf{x}} \log p_t(\mathbf{x})$, typically approximated by a trained score network, $\mathbf{s}_\theta(\mathbf{x}, t)$ or $\mathbf{s}_\theta(\mathbf{x}, \sigma)$ if parameterized by noise level $\sigma$.

The Elucidating Diffusion Models (EDM) framework (Karras et al., 2022; 2024a) parametrizes the diffusion process directly using the noise level $\sigma \in [\sigma_{\min}, \sigma_{\max}]$ rather than an abstract time $t$. This can be achieved by assuming $g(t) = \sqrt{2t}$ and $f(\mathbf{x}, t) = \mathbf{0}$, and using the initial condition $\mathbf{x}_T \sim \mathcal{N}(\mathbf{0}, \sigma_{\max}^2 \mathbf{I})$ for some fixed, sufficiently large $\sigma_{\max} > 0$. The perturbed density can be written as $p_t(\mathbf{x}) = p_\sigma(\mathbf{x}) = p(\mathbf{x}) * \mathcal{N}(\mathbf{x}; \mathbf{0}, \sigma^2 \mathbf{I})$.

The score network $\mathbf{s}_\theta(\mathbf{x}, \sigma)$ is trained via denoising score matching (Eq. 1). Sampling is done by solving the stochastic reverse diffusion SDE (Eq. 3) or the deterministic probability flow ODE (Eq. 4).

## 2.3 RANDOM UTILITY MODELS AND DENSITY ESTIMATION FROM CHOICE DATA

In the context of decision theory, the random utility model (RUM) represents the decision maker's stochastic utility $U$ as the sum of a deterministic utility and a stochastic perturbation (Train, 2009),

$$U(\mathbf{x}) = u(\mathbf{x}) + W(\mathbf{x}),$$

where $u : \mathcal{X} \to \mathbb{R}$ is a deterministic *utility function*, $W$ is a stochastic noise process, and the choice space $\mathcal{X}$ is a compact subset of $\mathbb{R}^d$. Given a set $\mathcal{C} \subset \mathcal{X}$ of possible alternatives, the decision maker selects an alternative $\mathbf{x} \in \mathcal{C}$ by solving the noisy utility maximization problem: $\mathbf{x} \sim \arg\max_{\mathbf{x}' \in \mathcal{C}} U(\mathbf{x}')$. Pairwise comparison is the most common form of choice data and corresponds to assuming that the choice set contains only two alternatives, $\mathcal{C} = \{\mathbf{x}, \mathbf{x}'\}$. The decision maker chooses $\mathbf{x}$ from $\mathcal{C}$, denoted by $\mathbf{x} \succ \mathbf{x}'$, if $u(\mathbf{x}) + w(\mathbf{x}) > u(\mathbf{x}') + w(\mathbf{x}')$ for a given realization $w$ of $W$. It is often assumed that $W$ is independent across $\mathbf{x}$, leading to so-called Fechnerian models (Becker et al., 1963), where the choice distribution conditional on $\mathcal{C}$ reduces to $F(u(\mathbf{x}) - u(\mathbf{x}'))$, with $F$ denoting the cumulative distribution function of $W(\mathbf{x}') - W(\mathbf{x})$.

Psychophysical experiments suggest that human perception of numerical magnitude follows a logarithmic scale (Dehaene, 2003). Assuming a RUM with utility function $u(\mathbf{x}) = \log p(\mathbf{x})$, the model's noise becomes additive in the log-transformed beliefs. In this paper, we consider two RUMs, explicitly including the noise level, as it is crucial for identifying $p(\mathbf{x})$[1]. First, we study the *generalized Bradley–Terry model* (Bradley & Terry, 1952) with $W \sim \text{Gumbel}(0, s)$, which induces the conditional choice distribution $F_{\text{Logistic}(0,s)}(u(\mathbf{x}) - u(\mathbf{x}'))$. Second, we consider the *exponential RUM* with $W \sim \text{Exp}(s)$, which yields a heavier-tailed conditional choice distribution $F_{\text{Laplace}(0,1/s)}(u(\mathbf{x}) - u(\mathbf{x}'))$.

Under these assumptions, the density estimation task is an instance of expert knowledge elicitation (O'Hagan, 2019; Mikkola et al., 2023), and it is closely related to (probabilistic) reward modeling (Leike et al., 2018; Dumoulin et al., 2024). Expert knowledge elicitation infers an expert's belief as a probability density $p(\mathbf{x})$ using only queries they can reliably answer, such as requests for specific quantiles or statistics of $p(\mathbf{x})$ (O'Hagan, 2019; Bockting et al., 2025) or comparisons like here. Recently, Dumoulin et al. (2024) reinterpreted reward modeling by referring to the target distribution as the "implicit preference distribution" and treating the reward as a probability distribution to be modeled.

## 3 BELIEF DENSITY AS A TEMPERED MARGINAL WINNER DENSITY

Let $p(\mathbf{x})$ be the expert's *belief density*. We assume the expert's choices follow a RUM with utility function $u(\mathbf{x}) = \log p(\mathbf{x})$. They observe two points independently drawn from the *sampling density* $\lambda(\mathbf{x})$, and the expert chooses one of the points. We denote the probability density of that point by $p_w(\mathbf{x})$ and refer to it as the *marginal winner density* (MWD). By marginalizing out the unobserved loser $\mathbf{x}'$ in a pairwise comparison where $\mathbf{x}$ is preferred ($\mathbf{x} \succ \mathbf{x}'$), $p_w(\mathbf{x})$ can be expressed as $2\lambda(\mathbf{x}) \int F(\log p(\mathbf{x}) - \log p(\mathbf{x}')) \lambda(\mathbf{x}') d\mathbf{x}'$ (Mikkola et al., 2024).

While Mikkola et al. (2024) empirically showed that $p(\mathbf{x})$ resembles a tempered version of $p_w(\mathbf{x})$ (i.e., $p(\mathbf{x}) \approx [p_w(\mathbf{x})]^\tau$ for some constant $\tau$), this relationship was not formally analyzed besides the theoretical limiting case of selecting the winner from infinitely many alternatives. In this section, we establish a more fundamental connection. We demonstrate that under two RUMs—the Bradley–Terry model and the exponential RUM—it is possible to find a tempering field $\tau(\mathbf{x})$ such that $\nabla \log p(\mathbf{x}) = \tau(\mathbf{x}) \nabla \log p_w(\mathbf{x})$ up to reparameterization of the space. This key relationship implies that, in principle, $p(\mathbf{x})$ **can be precisely recovered from $p_w(\mathbf{x})$ using score-based methods if $\tau(\mathbf{x})$ is known**. This finding motivates leveraging score-matching techniques for estimating the belief density. To analyze such score-based relationships and evaluate approximations, we use the Fisher divergence, which quantifies the difference between two distributions based on their scores:

$$F(p, q) = \int_{\mathcal{X}} \|\nabla \log p(\mathbf{x}) - \nabla \log q(\mathbf{x})\|^2 p(\mathbf{x}) d\mathbf{x}.$$

---

[1]Exact identification of $p(\mathbf{x})$ requires knowing both the correct noise family and noise level. We consider the noise to be known; see Appendix E.3 for validation of robustness to this choice.

## 3.1 Tempering Field

Consider a tempered probability density $p(\mathbf{x})$ constructed from another density $q(\mathbf{x})$ using a tempering constant $\tau > 0$:

$$p(\mathbf{x}) = \frac{q^\tau(\mathbf{x})}{\int_{\mathcal{X}} q^\tau(\mathbf{x}') d\mathbf{x}'}.$$

For such densities, the relationship between their scores is given by the product rule as

$$\nabla \log p(\mathbf{x}) = \tau \nabla \log q(\mathbf{x}) + \log q(\mathbf{x}) \nabla \tau = \tau \nabla \log q(\mathbf{x}).$$

The score of the tempered density becomes directly proportional to that of the original density, i.e., the two score vectors are collinear. Inspired by this relationship, we define a more general concept. We call a function $\tau : \mathcal{X} \to (0, \infty)$ a *tempering field* between $p$ and $q$ if their scores satisfy the following relation almost everywhere for $\mathbf{x} \in \mathcal{X}$:

$$\nabla \log p(\mathbf{x}) = \tau(\mathbf{x}) \nabla \log q(\mathbf{x}). \tag{5}$$

This implies the scores are collinear, with $\tau(\mathbf{x})$ as a position-dependent scaling. The tempering field thus describes a localized, score-level tempering.

## 3.2 Tempering Fields Under RUMs

With our theoretical framework in place, we analyze the relationship between the belief density $p$ and the MWD $p_w$ in terms of tempering fields for RUM models with utility $\log p$. We prove our main results for both the Bradley–Terry model and the exponential RUM, with $W \sim \text{Gumbel}(0, s)$ and $W \sim \text{Exp}(s)$. The treatment of the latter RUM is deferred to Appendix A.

To facilitate theoretical analysis, with no loss of generality, we assume a uniform sampling distribution $\lambda$ throughout this section, to remove the tilting of MWD $p_w(\mathbf{x})$. We then address a non-uniform $\lambda$ by reparameterizing the space so that it becomes uniform on a hypercube. The invariance of the RUM under the space reparameterizing is derived in Appendix D. The diffusion model is trained in the transformed space, and the generated samples are mapped back to the original space with the inverse transformation. We use the Rosenblatt transformation, which requires the conditional distribution functions of $\lambda$ (Rosenblatt, 1952), here assumed to be either known (e.g., when $\lambda$ is Gaussian) or numerically approximated. Other transformations, e.g. ones based on copulas or normalizing flows trained on samples from $\lambda$ (i.e., the combined data of winners and losers), could be used as well.

Under the following assumptions, a tempering field exists between the belief density and the MWD:

**Assumption 1.** $\text{supp}(p) \subseteq \text{supp}(\lambda)$.

**Assumption 2.** $\lambda$ is a uniform density over $\mathcal{X}$.

**Assumption 3.** $p$ is smooth, with $\nabla p \neq \mathbf{0}$ almost everywhere.

**Theorem 3.1.** *Assume $W \sim \text{Gumbel}(0, s)$. A tempering field $\tau(\mathbf{x})$ exists between the belief density $p$ and the MWD $p_w$, and it is given by the formula,*

$$\tau(\mathbf{x}) = s \left( \frac{\int_{\mathcal{X}} \frac{1}{1 + r_s(\mathbf{x}, \mathbf{x}')} d\mathbf{x}'}{\int_{\mathcal{X}} \frac{r_s(\mathbf{x}, \mathbf{x}')}{(1 + r_s(\mathbf{x}, \mathbf{x}'))^2} d\mathbf{x}'} \right), \tag{6}$$

*where $r_s(\mathbf{x}, \mathbf{x}') := p^{\frac{1}{s}}(\mathbf{x}') p^{-\frac{1}{s}}(\mathbf{x})$ is the $1/s$-tempered density ratio.*

*Proof.* Our constructive proof derives a scalar field that satisfies the defining relation. Specifically, for any fixed $\mathbf{x} \in \mathcal{X}$, direct manipulations yield a scalar $\tau(\mathbf{x}) > 0$ such that $\nabla_{\mathbf{x}} \log p(\mathbf{x}) - \tau(\mathbf{x}) \nabla_{\mathbf{x}} \log p_w(\mathbf{x}) = \mathbf{0}$. See Appendix B for the full proof. ☐

Fig. 2 illustrates the tempering field in Theorem 3.1 using $s = \sqrt{6/\pi^2}$ (unit variance noise). There exists a specific invariance relationship between the tempering and the noise scale. Specifically, if $\tau_{p,s}$ denotes the tempering field of RUM under the belief density $p$ and noise level $s > 0$, then by the tempering field theorems it is clear that for any exponent $\alpha > 0$: $\tau_{p^\alpha,s} = \frac{1}{s} \tau_{p^{\alpha s},1}$ and $\tau_{p^\alpha,s} = s \tau_{p^{\frac{\alpha}{s}},1}$, where the tempering fields are of the exponential RUM and the Bradley–Terry model, respectively.

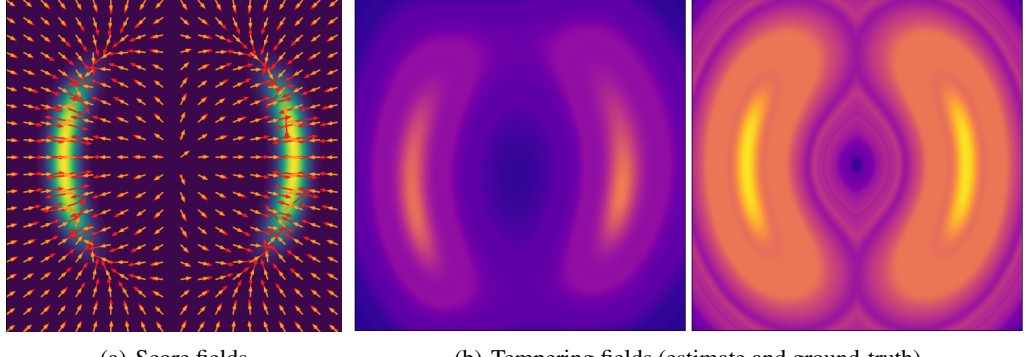

(a) Score fields        (b) Tempering fields (estimate and ground-truth)

Figure 2: Illustration of the relationship $\nabla \log p(\mathbf{x}) = \tau(\mathbf{x}) \nabla \log p_w(\mathbf{x})$ when $p$ is Twomoons2D (Stimper et al., 2022) and $\lambda$ is uniform. (a) The score of $p$ (red arrows) and the score of $p_w$ (orange arrows) under the Bradley–Terry model, scaled for better visualization. (b) The estimated tempering field $\tau(\mathbf{x})$ from 200 pairwise comparisons (left, Section 4.2) and the ground-truth (right, Theorem 3.1). Due to the collinearity of the scores, the red arrows equal the pointwise product of the orange arrows and the tempering field, which can be estimated (with an underestimation in this example).

## 3.3 ON THE CONSTANT TEMPERING APPROXIMATION

Even though our method directly estimates the full tempering field, our theory also sheds light on methods assuming constant tempering, such as Mikkola et al. (2024). It allows us to establish three quantities of interest related to approximating $p$ with a constant-tempered version of $q$: (i) the optimal constant tempering $\tau^\star > 0$ which minimizes $F(p, q^{\tau^\star})$, (ii) the approximation error $F(p, q^\tau)$ for any constant $\tau > 0$, and (iii) the approximation error $F(p, q^{\tau^\star})$ for the optimal constant tempering.

**Proposition 3.2.** *Assume that there exists a tempering field $\tau(\mathbf{x})$ between $p$ and $q$. The optimal tempering constant $\tau^\star = \arg\min_{\tau>0} F(p, q^\tau)$ can be written as,*

$$\tau^\star = \mathbb{E}_{X \sim p}\left(\omega(X)\tau(X)\right), \tag{7}$$

*where the stochastic weight $\omega \geq 0$ is given by $\omega(X) = \frac{\|\nabla \log q(X)\|^2}{\mathbb{E}_{X \sim p}\left(\|\nabla \log q(X)\|^2\right)}$.*

*Proof.* By the Leibniz integral rule,

$$\frac{\partial}{\partial \tau} F(p, q^\tau) = \int_{\mathcal{X}} 2 \left(\tau \|\nabla \log q(\mathbf{x})\|^2 - \langle \nabla \log q(\mathbf{x}), \nabla \log p(\mathbf{x})\rangle\right) p(\mathbf{x})d\mathbf{x}.$$

The divergence is quadratic in $\tau$ and the critical point is the global minimum, and by assumption $\langle \nabla \log q(\mathbf{x}), \nabla \log p(\mathbf{x})\rangle = \tau(\mathbf{x}) \|\nabla \log q(\mathbf{x})\|^2$. Algebraic manipulation gives the result. □

The approximation errors can be quantified in terms of the tempering field.

**Proposition 3.3.** *Let $\tau(\mathbf{x})$ be a tempering field. For any $\tau > 0$ it holds that*

$$F(p, q^\tau) = \mathbb{E}_{X \sim p}\left(|\tau - \tau(X)|^2 \|\nabla \log q(X)\|^2\right).$$

*Further, when $\tau^\star > 0$ is the optimal tempering, we have*

$$F(p, q^{\tau^\star}) = \mathbb{E}_{X \sim p}\left(\|\nabla \log q(X)\|^2 \tau^2(X)\right) - \frac{\left(\mathbb{E}_{X \sim p}\left(\tau(X) \|\nabla \log q(X)\|^2\right)\right)^2}{\mathbb{E}_{X \sim p}\left(\|\nabla \log q(X)\|^2\right)}.$$

*Proof.* See Appendix B. □

## 4    SCORE-BASED DENSITY ESTIMATION FROM PAIRWISE COMPARISONS

Building on Section 3, we now introduce our score-based density estimator for eliciting the belief density from pairwise comparisons. The method has two components. First, we train a diffusion model on the joint distribution of winners and losers using a masking scheme that ensures its marginal, that is MWD $p_w(\mathbf{x})$, can also be evaluated. Second, under the Bradley–Terry model, we provide a simple procedure to estimate the tempering field $\tau(\mathbf{x})$ and use it to de-temper the score-based estimate of the MWD. Details of both steps are explained next. The sampling distribution $\lambda(\mathbf{x})$ is assumed known, and we reparameterize the space to make it uniform, as explained in Section 3.2.

### 4.1    MODELING THE MWD

Our goal is to learn the perturbed score model of the MWD $\nabla \log[p_w(\mathbf{x}) * \mathcal{N}(\mathbf{x}; \mathbf{0}, \sigma^2 \mathbf{I})]$. We want to utilize all samples, both winners and losers. To do so we simultaneously learn the marginal $p_w(\mathbf{x}) = \int p_{\mathbf{x} \succ \mathbf{x}'}(\mathbf{x}, \mathbf{x}') d\mathbf{x}'$, and the full joint $p_{\mathbf{x} \succ \mathbf{x}'}(\mathbf{x}, \mathbf{x}')$ from the concatenated data of winners and losers. To learn the marginal, during training, half of the time we randomly mask $\mathbf{x}'$ and consider the score only with respect to $\mathbf{x}$.

We parametrize the score model as $\mathbf{s}_\theta(\mathbf{x}, \mathbf{x}', \sigma, \mathsf{joint}, \mathsf{temp})$, where $\mathsf{joint} \in \{0, 1\}$ and $\mathsf{temp} \in \{0, 1\}$ are conditioning flags: (a) When $\mathsf{joint} = 1$, the network takes both $\mathbf{x}$ and $\mathbf{x}'$ as input and models the score of the joint distribution, $\nabla \log p_{\mathbf{x} \succ \mathbf{x}'}(\mathbf{x}, \mathbf{x}')$. (b) When $\mathsf{joint} = 0$, the loser $\mathbf{x}'$ is masked (replaced with noise), and the network models the MWD score $\nabla \log p_w(\mathbf{x})$. The flag $\mathsf{temp}$ then controls whether the output is scaled by the tempering field: setting $\mathsf{temp} = 1$ yields an approximation to $\tau(\mathbf{x}) \nabla \log p_w(\mathbf{x}) = \nabla \log p(\mathbf{x})$.

This parametrization allows us to train a single score network on both winners and losers via denoising score matching (Eq. 1), while still enabling sampling from the belief density. During training, we randomly mask the loser with probability 0.5 (see Appendix C.1 for details). The MWD could also be estimated directly from winners alone, but this would ignore that losers carry information about where winners are less likely to be. We demonstrate the value of modeling the joint distribution empirically in Appendix C.1, while also confirming that we can accurately marginalize the joint model.

We stay as close as possible to the EDM-style diffusion model (Karras et al., 2024a). We use the perturbation kernel $p_\sigma(\tilde{\mathbf{x}} \mid \mathbf{x}) = \mathcal{N}(\tilde{\mathbf{x}}; \mathbf{x}, \sigma^2 \mathbf{I})$, which aligns with EDM and defines a forward diffusion process from $\sigma_{\min}$ to $\sigma_{\max}$, where $p_{\sigma_{\min}}(\mathbf{x}) \approx p(\mathbf{x})$ and $p_{\sigma_{\max}}(\mathbf{x}) \approx \mathcal{N}(\mathbf{0}, \sigma_{\max}^2 \mathbf{I})$. A detailed description is provided in Appendix C.2. Algorithm 1 summarizes the method.

### 4.2    TEMPERING FIELD ESTIMATION

The tempering field under the Bradley–Terry model (Eq. 6) has a particularly convenient form as it depends only on the *belief density ratio* $r(\mathbf{x}, \mathbf{x}') := p(\mathbf{x}')/p(\mathbf{x})$ and the RUM noise level $s > 0$. Note that this ratio is different from what the phrase *density ratio* often refers to; this is the ratio of the same density for two inputs, not a ratio of two densities for the same $\mathbf{x}$. It does not depend on the normalizing constant of the belief density and is hence straightforward to estimate via maximum-likelihood estimation (MLE), assuming careful regularization. We train a simple estimator for $r(\mathbf{x}, \mathbf{x}')$ by maximizing the Bradley–Terry model likelihood of the pairwise comparison data. If we parametrize the density ratio (or its logarithm) as a neural network $r_\theta$, the parameters $\theta$ can be optimized by minimizing the loss $\mathcal{L}(\theta) \propto \mathrm{Softplus}(\log r_\theta(\mathbf{x}, \mathbf{x}')/s)$, where $\mathbf{x}$ and $\mathbf{x}'$ are winner and loser points, respectively. The Softplus loss comes from the assumption of $W \sim \mathrm{Gumbel}(0, s)$.

We plug the learned $r_\theta$ into the integrals in Eq. 6, where the integrals are computed using importance sampling with the MWD model acting as the importance sampler. The resulting plug-in Monte Carlo estimator of the tempering field is consistent but biased. Similar biased ratio estimators have been used in self-normalized importance sampling (Owen, 2013) and in every-visit off-policy value estimation in reinforcement learning (Sutton et al., 1998) due to favorable variance properties. See Appendix C.5 for more discussion on the estimator. For details on evaluating the importance weights, which correspond to the (reciprocal of) density of the MWD diffusion model, see Appendix C.7. Algorithm 2 summarizes the estimation procedure.

## 4.3 Belief density sampling

Given the perturbed MWD score network $\mathbf{s}_\theta(\mathbf{x}, \sigma) \approx \nabla \log[p_w(\mathbf{x}) * \mathcal{N}(\mathbf{x}; \mathbf{0}, \sigma^2\mathbf{I})]$ and the estimate of the tempering field $\tau(\mathbf{x})$, we can draw approximate samples from the belief density $p(\mathbf{x})$ using the score-scaled ALD. Specifically, we iteratively run Eq. 2 with the score $\tau(\mathbf{x})\mathbf{s}_\theta(\mathbf{x}, \sigma)$. The tempering field relation (Eq. 5) shows that for $\sigma = 0$ this procedure would be equivalent to sampling from $p(\mathbf{x})$ and ALD is theoretically valid at the small-noise limit (Welling & Teh, 2011), making it an appealing choice as a sampling algorithm. However, for $\sigma > 0$ the algorithm is not exact. We show that it works empirically well (Section 5), but characterization of the approximation error remains as future work.

---

**Algorithm 1** Full algorithm

**require:** choice data $\mathcal{D} = \{[\mathbf{x}_i, \mathbf{x}_i'] \mid \mathbf{x}_i \succ \mathbf{x}_i'\}_{i=1}^n$
**output:** samples from the belief density or a trained diffusion model for it it
train $\mathbf{s}_\theta(\mathbf{x}, \mathbf{x}', \sigma, \mathsf{joint})$ using DSM (Eq. 1) on $\mathcal{D}$
  50% prob.: set $\mathsf{joint} = 0$ and mask $\mathbf{x}'$ with $\mathcal{N}(\mathbf{0}, \sigma_t^2\mathbf{I})$
  50% prob.: set $\sigma$ to noise schedule $(\sigma_t)_{t=1}^L$
initialize $\tau(\mathbf{x})$ given $\mathcal{D}$ and $\mathbf{s}_\theta$
sample $\mathcal{D}^\star$ using $\tau(\mathbf{x})$-scaled ALD with the score $\mathbf{s}_\theta(\mathbf{x}, \mathbf{x}', (\sigma_t)_{t=1}^L, \mathsf{joint}=0)$
——————————— *optional* ———————————
train $\mathbf{s}_{\theta_{\text{MWD}}}(\mathbf{x}, \sigma, \mathsf{temp})$ using DSM on $\mathcal{D}^\star$

---

**return:** $\mathcal{D}^\star$ or $\mathbf{s}_{\theta_{\text{MWD}}}(\mathbf{x}, \sigma, \mathsf{temp}=1)$

---

**Algorithm 2** $\tau(\mathbf{x})$

**require:** noise level s, $\mathcal{D}$, $\mathbf{s}_\theta$
**initialize:**
  train $r_\theta(\mathbf{x}, \mathbf{x}') \approx \frac{p(\mathbf{x}')}{p(\mathbf{x})}$ as MLE of $\mathcal{D}$ (Softplus loss)
  sample $m$ points $\mathbf{X}$ with densities $p_w(\mathbf{X})$ using $\mathbf{s}_\theta$
**input:** $\mathbf{x}$
$\mathbf{r} = (r_\theta(\mathbf{x}, \mathbf{X}))^{\frac{1}{s}}$
**return:**
$s\left(\frac{\sum_{i=1}^m \frac{1}{1+\mathbf{r}_i} \frac{1}{p_w(\mathbf{X}_i)}}{\sum_{i=1}^m \frac{\mathbf{r}_i}{(1+\mathbf{r}_i)^2} \frac{1}{p_w(\mathbf{X}_i)}}\right)$

---

## 5 Experiments

We evaluate the method on synthetic data generated from a RUM. We then consider an experiment where a large language model (LLM) serves as a proxy for a human expert, demonstrating the method's applicability in settings where data does not follow a RUM model. Our experimental setup closely follows that of Mikkola et al. (2024), with the key distinction that we only query pairwise comparisons, not considering the easier case of ranking multiple candidates. We empirically compare against their flow-based model, using their implementation, as the only previous method for the task. We consider two variants of our *score-based method*: *score–$\tau(\mathbf{x})$*, which uses the full tempering field (Section 4), and *score–$\tau^\star$*, which uses a constant tempering estimated via Proposition 3.2. This allows direct quantification of the importance of modeling the whole field.

**Setup and evaluation** For a $d$-dimensional target, we query $1000d$ pairwise comparisons to ensure reliable comparison between the methods but remaining well below the large-sample regime typical for the diffusion model literature. For $d \leq 4$, the sampling distribution $\lambda$ is uniform. For $d > 4$, $\lambda$ is a diagonal Gaussian (Gaussian mixture in Mixturegaussians10D) centered at the target mean, with a variance three times that of the target's.[2] The simulated expert follows the Bradley–Terry model with utility given by $\log p$, where the belief density $p$ varies in dimensionality, modality, and detail. We set the noise level $s = \sqrt{6/\pi^2}$ (unit variance). As the diffusion model, we adopt an EDM-style architecture with a MLP score network, implemented on top of Karras et al. (2024b). For further details, see Appendix C and E. We assess performance qualitatively by visually comparing $2D$ and $1D$ marginals of the target density and the estimate, and quantitatively using two metrics: the Wasserstein distance and the mean marginal total variation distance (MMTV; Acerbi, 2020). Results are reported as means and standard deviations over replicate runs.

**Experiment 1: Synthetic low dimensional targets with uniform sampling** We consider low-dimensional synthetic target densities that may exhibit non-trivial geometry or multimodality. The set includes five target distributions, see Appendix E.1. Table 1 (top) shows that the score-based method is clearly superior for all targets, with at least $50\%$ (Wasserstein) and $25\%$ (MMTV) reduction of error

---

[2]For non-uniform sampling distributions, we transform the space with a Rosenblatt transform such that the sampling pdf becomes uniform, with an appropriate Jacobian transformation to the densities.

Table 1: Density estimation from pairwise comparisons: *score–τ*$^\star$ and *score–τ*(**x**) denote our methods with constant and varying tempering fields, respectively, and *flow* refers to (Mikkola et al., 2024). Bold indicates the best method, and underline indicates results that are not significantly worse (paired two-sided Wilcoxon signed-rank test, $p > 0.05$).

| $p(\mathbf{x})$ | wasserstein (↓) | | | MMTV (↓) | | |
|---|---|---|---|---|---|---|
| | *flow* | *score–τ*$^\star$ | *score–τ*(**x**) | *flow* | *score–τ*$^\star$ | *score–τ*(**x**) |
| Onemoon2D | 1.37 (±0.03) | 0.44 (±0.13) | **0.37** (±0.14) | 0.54 (±0.00) | 0.25 (±0.06) | **0.22** (±0.06) |
| Twomoons2D | 1.29 (±0.06) | 0.54 (±0.14) | **0.44** (±0.09) | 0.53 (±0.01) | 0.15 (±0.03) | **0.14** (±0.02) |
| Ring2D | 0.87 (±0.03) | 0.40 (±0.07) | **0.39** (±0.08) | 0.40 (±0.01) | 0.27 (±0.01) | **0.26** (±0.01) |
| Gaussian4D | 6.12 (±0.05) | 1.69 (±0.22) | **1.40** (±0.24) | 0.72 (±0.01) | **0.41** (±0.05) | 0.44 (±0.08) |
| Mixturegaussians4D | 3.75 (±0.02) | 1.23 (±0.06) | **1.09** (±0.09) | 0.53 (±0.01) | 0.27 (±0.01) | **0.22** (±0.02) |
| Stargaussian6D | 2.25 (±0.02) | 1.55 (±0.04) | **1.28** (±0.04) | 0.19 (±0.00) | 0.18 (±0.02) | **0.16** (±0.01) |
| Mixturegaussians10D | 1.41 (±0.01) | **1.10** (±0.12) | 1.33 (±0.11) | 0.19 (±0.00) | **0.14** (±0.02) | 0.26 (±0.06) |
| Gaussian16D | 5.50 (±0.03) | 5.00 (±0.07) | **5.00** (±0.06) | 0.16 (±0.00) | 0.13 (±0.00) | **0.13** (±0.00) |

compared to the flow method. In most cases, using the full tempering field $\tau(\mathbf{x})$ is better than relying on the best constant tempering, but we outperform the flow-based method even when restricted to constant tempering as in their approach. Visual inspection (Fig. 3 (a-b) and Figs. F.1-F.3) confirms this. The flow method captures the density relatively well but clearly overestimates the low-density regions, whereas our estimate is essentially perfect here.

**Experiment 2: Synthetic targets with Gaussian sampling**    For higher-dimensional experiments we replace uniform sampling with more concentrated Gaussian sampling, since otherwise the probability that both **x** and **x**′ fall in low-density regions increases dramatically, making it impossible to learn $p(\mathbf{x})$ well. The set includes three target distributions, see Appendix E.1. Table 1 (bottom) shows that we again outperform the flow-based method in Wasserstein distance but in terms of MMTV the methods are closer. Visually, the score-based method usually gives sharper and better estimates (*e.g.*, compare Fig. F.6 and F.7), but suffers in terms of the MMTV metric due to occasional too-tight marginals resulting from overestimating the tempering field (*e.g.*, Fig. F.8).

**Experiment 3: LLM as a proxy for the expert**    To illustrate the method in a real belief density estimation task without user experiments, we replicate the LLM experiment of Mikkola et al. (2024), except that we query 220 pairwise comparisons instead of 5-wise rankings. Using the data from (Mikkola et al., 2024), we uniformly sample 220 pairwise comparisons across all eight features and prompt the LLM for belief judgments. The LLM[3] acts as a proxy for a human expert, providing its belief about what houses in California are like, restricted to the features available in the California housing dataset (Pace & Barry, 1997). The LLM's belief is inferred solely from the pairwise queries, without providing the LLM any direct access to the data itself.

This experiment leverages the finding that LLMs learn statistical features from the vast training data and can be queried about it (Brown et al., 2020; Requeima et al., 2024). While the aim is to infer the belief density of the LLM—which is unknown—our main goal here is to validate our method, rather than to analyze the beliefs of this particular LLM. We do this by comparing the belief estimate with the empirical data distribution: Similarity between the two suggests the elicitation method yields a reasonable belief estimate, and differences can be interpreted as possible biases the LLM might have. Fig. 3 (c) shows clear distributional similarities; e.g. the marginals of *AveRooms* and *MedInc* exhibit similar shapes. See Appendix F.2 for complete results, with Table F.1 quantifying the accuracy.

## 6    DISCUSSION

We proved the theoretical connection for two common RUMs but we believe it extends to other RUMs as well, although a closed-form tempering field is not guaranteed e.g. for the Thurstone–Mosteller model (Thurstone, 1927) where the choice probability requires integration.

The difficulty of the belief estimation problem, naturally, depends heavily on the sampling distribution $\lambda(\mathbf{x})$. For example, when the support of $p(\mathbf{x})$ is much smaller than that of $\lambda(\mathbf{x})$, it becomes

---

[3]For a direct comparison with Mikkola et al. (2024), we also use Claude 3 Haiku by Anthropic (2024).

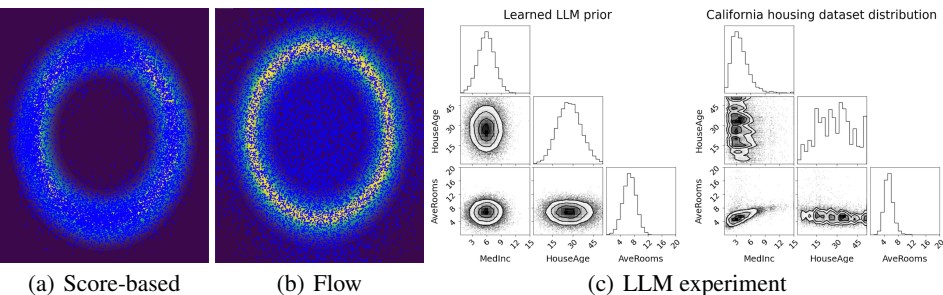

(a) Score-based      (b) Flow          (c) LLM experiment

Figure 3: (a-b) Samples from *score-based* and *flow* estimates of Ring2D, with contours indicating the true density. Ring2D illustrates an extreme case where the score-based method clearly outperforms the flow method: the flow model oversamples the center of the ring, where the MWD also has moderate density, whereas the score-based method can downweight it using the tempering field. (c) Cross-plot of the first three variables in the LLM expert elicitation experiment. Full cross-plot and comparison to the flow method are shown in Figs. F.10 and F.11. The score-based method tends to generate Gaussian-like marginals in this extremely limited data setting.

nearly impossible to obtain sharp estimates of $p(\mathbf{x})$, and the problem is further exacerbated in high-dimensional spaces. For this reason, we see potential in active learning methods that concentrate sampling in high-density regions of $p(\mathbf{x})$. In this work, we assume that $\lambda(\mathbf{x})$ is given as it would be in many applications (e.g. the elicitation protocol in prior elicitation), but for active learning or learning beliefs from public preference data, an additional density estimation step is required to learn $\lambda(\mathbf{x})$.

We demonstrated that low-dimensional targets can be learned from a few hundreds of pairwise comparisons (Fig. 1). However, in extremely limited data regimes, say below $100d$ pairwise comparisons, the robustness of our method is not guaranteed without carefully tuning hyperparameters such as those of the diffusion model—a class of models that are notoriously difficult to train in a stable manner (Karras et al., 2024b, Appendix B.5). Stability could likely be improved by adopting best practices from the field (Karras et al., 2022) and incorporating recent advances in learning score models from limited data (Li et al., 2024). Similarly, the tempering field estimate (Section 4.2) depends on the density ratio estimate $r_\theta$, which is sensitive to the network's regularization. Misspecified values of the $\ell_2$ regularization for the network weights $\theta$ or the RUM noise level $s$ will lead to under- or overestimation of the tempering field, due to MLE struggling to determine the global scale and tails.

While our method is theoretically grounded and clearly outperforms the flow-based method of Mikkola et al. (2024) in estimation accuracy, it has some limitations. The flow allows faster sampling, only requiring a single forward pass, and also more efficient and stable evaluation of the density. Our method requires numerically solving the probability-flow ODE for evaluating the density, and it is still open whether diffusion models provide reliable pointwise density estimates (Zheng et al., 2023).

The connection between the target density $p$ and the MWD $p_w$ may have applications beyond expert knowledge elicitation, such as fine-tuning generative models (Wallace et al., 2024) using pairwise data, a perspective explored by Dumoulin et al. (2024). Specifically, when $\lambda(\mathbf{x})$ represents a pretrained generative model conditioned on a prompt $c$, our theory suggests that training the MWD and tempering field on individual-level data (rather than pooled data) yields a probabilistic reward model $\text{reward}(c, \mathbf{x}) = p(\mathbf{x}|c)$. This conditional tempered MWD captures the distribution of samples (e.g., images) aligned with prompt $c$ from that specific individual's perspective.

## 7 CONCLUSIONS

Despite two decades of active research on learning from preference data, following the pioneering works of Chu & Ghahramani (2005); Fürnkranz & Hüllermeier (2010), the question of how to learn flexible *densities* from such data has remained elusive. We established the missing theoretical basis by showing how the score of the target density relates to quantities that can be estimated, enabling the use of powerful modern density estimators for this task. Our approach demonstrates superior performance over recent flow-based solutions in representing human beliefs.

ACKNOWLEDGMENTS

The authors acknowledge the research environment provided by ELLIS Institute Finland. The work was supported by the Research Council of Finland Flagship programme: Finnish Center for Artificial Intelligence FCAI, and additionally by the grants 363317 and 358980. The authors acknowledge support from CSC – IT Center for Science, Finland, for computational resources.

**Reproducibility statement** The source code for reproducing the experiments is available at https://github.com/petrus-mikkola/pairwise2diffusion. Section 4 describes the method and its implementation, while Appendix C details the specific components and training settings required to replicate the experiments. Appendix B provides the full proofs of the theoretical results presented in the main text.

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

## APPENDIX

This appendix provides additional technical material complementing the main paper. Appendix A presents extended theoretical results for the exponential noise RUM. Appendix B contains the proofs. Appendix C provides method details and implementation specifications. Appendix D shows the RUM invariance under space reparameterization to a uniform distribution. Appendix E reports experimental details, the runtime breakdown, and additional ablations on RUM model misspecification. Appendix F includes plots of the learned belief densities and a description of the LLM experiment. Appendix G describes the use of LLMs in the preparation of this paper.

## LIST OF NOTATION

| | |
|---|---|
| $\mathbf{x}$ | A vector |
| $\mathbf{X}$ | A matrix |
| $\mathbf{x}_i$ | Element $i$ of vector $\mathbf{x}$ or $i^{th}$ observation |
| $X$ | A vector-valued random variable |
| $W$ | RUM noise, a scalar random variable or a stochastic process |
| $W(\mathbf{x})$ | A stochastic process at index $\mathbf{x}$ |
| $\succ$ | A binary strict preference relation |
| $\mathbf{x} \succ \mathbf{x}'$ | $\mathbf{x}$ is chosen over $\mathbf{x}'$, where $\mathbf{x}$ is winner point and $\mathbf{x}'$ is loser point |
| $p(\mathbf{x})$ | A probability density function evaluated at $\mathbf{x}$ |
| $p$ | Belief density |
| $\lambda$ | Sampling density |
| $p_w$ | Marginal winner density (MWD) |
| $p_{\mathbf{x} \succ \mathbf{x}'}$ | Joint density of winner-loser pairs encoded in the order of the arguments |
| $u$ | Utility function |
| $\nabla f(\mathbf{x})$ | The gradient of $f$ with respect to $\mathbf{x}$, and evaluated at $\mathbf{x}$ |
| $\mathbf{s}_\theta$ | Noise-conditioned score network with parameters $\theta$ |
| $\mathbf{s}_\theta(\mathbf{x}, \mathbf{x}', \sigma)$ | Noise-conditioned score network evaluated at winner-loser pair $(\mathbf{x}, \mathbf{x}')$ and noise scale $\sigma$ |
| $s$ | RUM noise level, a positive scalar |
| $\tau$ | Tempering field, a scalar field $\tau : \mathcal{X} \to (0, \infty)$ or a positive scalar (in case of constant field) |
| $\tau(\mathbf{x})$ | Tempering field evaluated at $\mathbf{x}$, a positive scalar |
| $\tau^\star$ | Optimal tempering constant, a positive scalar |
| $\omega$ | Weighting function, a scalar field $\omega : \mathcal{X} \to [0, \infty)$ |
| $\omega(X)$ | Stochastic weight, a scalar random variable |
| $\|\cdot\|$ | $\ell_2$-norm (Euclidean norm) |

## A    TEMPERING FIELD UNDER THE EXPONENTIAL NOISE RUM

In this appendix, we state the existence and provide a closed-form expression for the tempering field when the expert's choice model follows the exponential RUM with $W \sim Exp(s)$.

**Theorem A.1.** *Assume $W \sim Exp(s)$. A tempering field $\tau(\mathbf{x})$ exists between the belief density $p$ and the MWD $p_w$, and it is given by the formula*

$$\tau(\mathbf{x}) = \frac{1}{s}\left(\frac{2vol(L_{\mathbf{x}}) + 2p^s(\mathbf{x})\int_{U_{\mathbf{x}}} p^{-s}(\mathbf{x}')d\mathbf{x}'}{p^s(\mathbf{x})\int_{U_{\mathbf{x}}} p^{-s}(\mathbf{x}')d\mathbf{x}' + p^{-s}(\mathbf{x})\int_{L_{\mathbf{x}}} p^s(\mathbf{x}')d\mathbf{x}'} - 1\right), \tag{A.1}$$

*where the sublevel set $L_{\mathbf{x}} = \{\mathbf{x}' \in \mathcal{X} \mid p(\mathbf{x}) \geq p(\mathbf{x}')\}$ and the superlevel set $U_{\mathbf{x}} = \mathcal{X} \setminus L_{\mathbf{x}}$.*

*Proof Sketch.* For a fixed $\mathbf{x} \in \mathcal{X}$, after lengthy manipulations, we get $\tau(\mathbf{x}) > 0$ such that $\nabla_{\mathbf{x}} \log p(\mathbf{x}) - \tau(\mathbf{x})\nabla_{\mathbf{x}} \log p_w(\mathbf{x}) = \mathbf{0}$. To handle the change in integration domains, we apply the generalized Leibniz integral rule (Flanders, 1973). See Appendix B for the full proof. □

Fig. A.1 illustrates the tempering fields from Theorem A.1 and Theorem 3.1, using $s = \sqrt{6/\pi^2}$ for Bradley–Terry and $s = 1$ for exponential RUM, both resulting in unit variance for ease of comparison. The tempering fields are extremely similar, but tempering in high-density regions appears slightly more pronounced in the Bradley–Terry model, due to a lighter-tailed conditional choice distribution.

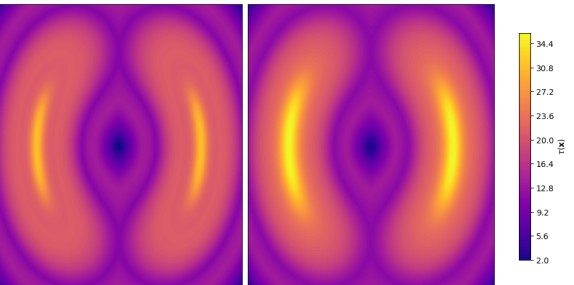

Figure A.1: Illustration of the tempering fields under two different RUMs when $p$ is Twomoons2D (Stimper et al., 2022). The tempering field $\tau(\mathbf{x})$ of the exponential RUM (left, Theorem A.1) and the Bradley–Terry model (right, Theorem 3.1).

## B    PROOFS

**Theorem A.1.** *Assume $W \sim Exp(s)$. A tempering field $\tau(\mathbf{x})$ exists between the belief density $p$ and the MWD $p_w$, and it is given by the formula,*

$$\tau(\mathbf{x}) = \frac{1}{s}\left(\frac{2vol(L_{\mathbf{x}}) + 2p^s(\mathbf{x})\int_{U_{\mathbf{x}}} p^{-s}(\mathbf{x}')d\mathbf{x}'}{p^s(\mathbf{x})\int_{U_{\mathbf{x}}} p^{-s}(\mathbf{x}')d\mathbf{x}' + p^{-s}(\mathbf{x})\int_{L_{\mathbf{x}}} p^s(\mathbf{x}')d\mathbf{x}'} - 1\right),$$

*where the sublevel set $L_{\mathbf{x}} = \{\mathbf{x}' \in \mathcal{X} \mid p(\mathbf{x}) \geq p(\mathbf{x}')\}$ and the superlevel set $U_{\mathbf{x}} = \{\mathbf{x}' \in \mathcal{X} \mid p(\mathbf{x}) < p(\mathbf{x}')\}$.*

*Proof.* We want to show that for each $\mathbf{x} \in \mathcal{X}$, there exists a scalar $\tau(\mathbf{x}) > 0$ such that $\nabla \log p(\mathbf{x}) - \tau(\mathbf{x})\nabla \log p_w(\mathbf{x}) = \mathbf{0}$. Our constructive proof determines this scalar through brute-force calculation. Fix a point $\mathbf{x} \in \mathcal{X}$, and denote a constant $\tau(\mathbf{x}) = \tau > 0$.

Under the exponential RUM, the MWD $p_w(\mathbf{x})$ equals to

$$p_w(\mathbf{x}) = 2\lambda(\mathbf{x})\int_{\mathcal{X}} F_{\text{Laplace}(0,1/s)}\left(\log p(\mathbf{x}) - \log p(\mathbf{x}')\right)\lambda(\mathbf{x}')d\mathbf{x}'. \tag{B.1}$$

For uniform $\lambda$, this implies

$$\nabla_{\mathbf{x}} \log p_w(\mathbf{x}) = \nabla_{\mathbf{x}} \log \int_{\mathcal{X}} F_{\text{Laplace}(0,1/s)} \left(\log p(\mathbf{x}) - \log p(\mathbf{x}')\right) d\mathbf{x}'. \tag{B.2}$$

Let $L_{\mathbf{x}}, U_{\mathbf{x}} \subset \mathcal{X}$ be the regions $L_{\mathbf{x}} = \{\mathbf{x}' \in \mathcal{X} \mid p(\mathbf{x}) \geq p(\mathbf{x}')\}$ and $U_{\mathbf{x}} = \{\mathbf{x}' \in \mathcal{X} \mid p(\mathbf{x}) < p(\mathbf{x}')\}$. Straightforward manipulations give us,

$$\tau \nabla_{\mathbf{x}} \log p_w(\mathbf{x}) - \nabla_{\mathbf{x}} \log p(\mathbf{x}) =$$

$$= \tau \nabla_{\mathbf{x}} \log \left( \int_{U_{\mathbf{x}}} \frac{1}{2} p^s(\mathbf{x}) p^{-s}(\mathbf{x}') d\mathbf{x}' + \int_{L_{\mathbf{x}}} \left(1 - \frac{1}{2} p^{-s}(\mathbf{x}) p^s(\mathbf{x}')\right) d\mathbf{x}' \right) - \tau \nabla_{\mathbf{x}} \log p^{\frac{1}{\tau}}(\mathbf{x})$$

$$= \tau \nabla_{\mathbf{x}} \log \frac{vol(L_{\mathbf{x}}) + \frac{1}{2} p^s(\mathbf{x}) \int_{U_{\mathbf{x}}} p^{-s}(\mathbf{x}') d\mathbf{x}' - \frac{1}{2} p^{-s}(\mathbf{x}) \int_{L_{\mathbf{x}}} p^s(\mathbf{x}') d\mathbf{x}'}{p^{\frac{1}{\tau}}(\mathbf{x})}.$$

Because $\nabla_{\mathbf{x}} \log f(\mathbf{x}) = \nabla_{\mathbf{x}} f(\mathbf{x}) / f(\mathbf{x})$, the above vanishes, if and only if,

$$\nabla_{\mathbf{x}} \left( p^{-\frac{1}{\tau}}(\mathbf{x}) vol(L_{\mathbf{x}}) + \frac{1}{2} p^{s - \frac{1}{\tau}}(\mathbf{x}) \int_{U_{\mathbf{x}}} p^{-s}(\mathbf{x}') d\mathbf{x}' - \frac{1}{2} p^{-s - \frac{1}{\tau}}(\mathbf{x}) \int_{L_{\mathbf{x}}} p^s(\mathbf{x}') d\mathbf{x}' \right) = \mathbf{0}.$$

We will apply to LHS the generalized Leibniz integral rule (Flanders, 1973) for each fixed dimension $j \in \{1, ..., d\}$ by interpreting $\frac{\partial}{\partial \mathbf{x}_j}$ as differentiation with respect to the time. To justify the generalized Leibniz integral rule, note that the boundaries $\partial L_{\mathbf{x}} = \partial U_{\mathbf{x}}$ are defined by an implicit function $g : \mathcal{X} \to \mathcal{X}$ whose graph is the set $\{(\mathbf{x}, g(\mathbf{x}))\} = \{(\mathbf{x}, \mathbf{x}') \in \mathcal{X}^2 \mid f(\mathbf{x}, \mathbf{x}') = 0\}$, where the function $f(\mathbf{x}, \mathbf{x}') := p(\mathbf{x}) - p(\mathbf{x}')$ is continuously differentiable by Assumption 3. Moreover, since $\nabla_{\mathbf{x}'} f(\mathbf{x}, \mathbf{x}') = -\nabla p(\mathbf{x}') \neq 0$ almost everywhere, the implicit function theorem implies that the level set $\{(\mathbf{x}, \mathbf{x}') \mid f(\mathbf{x}, \mathbf{x}') = 0\}$ locally defines $\mathbf{x}'$ as a differentiable function of $\mathbf{x}$ almost everywhere. Therefore, $g(\mathbf{x})$ is continuously differentiable almost everywhere.

For a smooth function $f : \mathcal{X} \to \mathbb{R}_+$ consider,

$$\frac{\partial}{\partial \mathbf{x}_j} \int_{A_{\mathbf{x}}} f(\mathbf{x}') d\mathbf{x}',$$

where $A_{\mathbf{x}} = L_{\mathbf{x}}$ or $A_{\mathbf{x}} = U_{\mathbf{x}}$. Interpret the scalar $\mathbf{x}_j$ as time, and apply the generalized Leibniz integral rule,

$$\frac{\partial}{\partial \mathbf{x}_j} \int_{A_{\mathbf{x}}} f(\mathbf{x}') d\mathbf{x}' = \int_{A_{\mathbf{x}}} \frac{\partial}{\partial \mathbf{x}_j} f(\mathbf{x}') d\mathbf{x}' + \int_{\partial A_{\mathbf{x}}} f(\mathbf{x}') (\mathbf{n} \cdot \mathbf{v}) dS,$$

where $\mathbf{n}$ is the unit normal vector pointing outwards from the boundary $\partial A_{\mathbf{x}}$, $\mathbf{v}$ is the Eulerian velocity of the boundary $\partial A_{\mathbf{x}}$ when $\mathbf{x}_j$ is interpreted as time, and $dS$ is the surface element. The first term in RHS vanishes. For the second term, consider the level set $\{\mathbf{x}' \in \mathcal{X} \mid p(\mathbf{x}) - p(\mathbf{x}') = 0\}$. The normal vector is orthogonal to this level set, which equals to gradient with respect to $\mathbf{x}'$, $\mathbf{n} = \nabla_{\mathbf{x}'} (p(\mathbf{x}) - p(\mathbf{x}')) / \|\nabla_{\mathbf{x}'} (p(\mathbf{x}) - p(\mathbf{x}'))\| = -\nabla_{\mathbf{x}'} p(\mathbf{x}') / \|\nabla_{\mathbf{x}'} p(\mathbf{x}')\|$. This corresponds to the normal vector of $L_{\mathbf{x}}$ while the normal vector of $U_{\mathbf{x}}$ is with the opposite sign.

Let us consider $\mathbf{v} = (\frac{\partial \mathbf{x}'_1}{\partial \mathbf{x}_j}, ..., \frac{\partial \mathbf{x}'_N}{\partial \mathbf{x}_j})$, the velocity of the boundary with respect to $\mathbf{x}_j$. The total derivative of the boundary should not change,

$$D_{\mathbf{x}_j}(p(\mathbf{x}) - p(\mathbf{x}')) = 0$$

$$\frac{\partial}{\partial \mathbf{x}_j}(p(\mathbf{x}) - p(\mathbf{x}')) + \sum_{i=1}^d \frac{\partial}{\partial \mathbf{x}'_i}(p(\mathbf{x}) - p(\mathbf{x}')) \frac{\partial \mathbf{x}'_i}{\partial \mathbf{x}_j} = 0$$

$$\frac{\partial}{\partial \mathbf{x}_j} p(\mathbf{x}) - \sum_{i=1}^d \frac{\partial}{\partial \mathbf{x}'_i} p(\mathbf{x}') \frac{\partial \mathbf{x}'_i}{\partial \mathbf{x}_j} = 0.$$

Taking the dot product of the constraint with the normal vector gives,

$$\mathbf{n} \cdot \mathbf{v} = -\frac{\frac{\partial}{\partial \mathbf{x}_j} p(\mathbf{x})}{\|\nabla_{\mathbf{x}'} p(\mathbf{x}')\|}.$$

Since $p(\mathbf{x}') = p(\mathbf{x})$ on $\partial L_{\mathbf{x}} = \partial U_{\mathbf{x}}$,

$$\frac{\partial}{\partial \mathbf{x}_j} \int_{L_{\mathbf{x}}} p^s(\mathbf{x}') d\mathbf{x}' = -p^s(\mathbf{x}) \frac{\partial}{\partial \mathbf{x}_j} p(\mathbf{x}) \int_{\partial L_{\mathbf{x}}} \frac{1}{\|\nabla_{\mathbf{x}'} p(\mathbf{x}')\|} dS(\mathbf{x}')$$

$$\frac{\partial}{\partial \mathbf{x}_j} \int_{U_{\mathbf{x}}} p^{-s}(\mathbf{x}') d\mathbf{x}' = p^{-s}(\mathbf{x}) \frac{\partial}{\partial \mathbf{x}_j} p(\mathbf{x}) \int_{\partial L_{\mathbf{x}}} \frac{1}{\|\nabla_{\mathbf{x}'} p(\mathbf{x}')\|} dS(\mathbf{x}')$$

$$\frac{\partial}{\partial \mathbf{x}_j} \int_{L_{\mathbf{x}}} d\mathbf{x}' = -\frac{\partial}{\partial \mathbf{x}_j} p(\mathbf{x}) \int_{\partial L_{\mathbf{x}}} \frac{1}{\|\nabla_{\mathbf{x}'} p(\mathbf{x}')\|} dS(\mathbf{x}').$$

Together these imply that,

$$\left( p^{-\frac{1}{\tau}}(\mathbf{x}) \frac{\partial}{\partial \mathbf{x}_j} \int_{L_{\mathbf{x}}} d\mathbf{x}' + \frac{1}{2} p^{s-\frac{1}{\tau}}(\mathbf{x}) \frac{\partial}{\partial \mathbf{x}_j} \int_{U_{\mathbf{x}}} p^{-s}(\mathbf{x}') d\mathbf{x}' - \frac{1}{2} p^{-s-\frac{1}{\tau}}(\mathbf{x}) \frac{\partial}{\partial \mathbf{x}_j} \int_{L_{\mathbf{x}}} p^s(\mathbf{x}') d\mathbf{x}' \right) = 0.$$

We are left with the following terms that vanish,

$$p^{-\frac{1}{\tau}-1}(\mathbf{x}) \nabla_{\mathbf{x}} p(\mathbf{x}) \left( -\frac{1}{\tau} vol(L_{\mathbf{x}}) + \frac{s-\frac{1}{\tau}}{2} p^s(\mathbf{x}) \int_{U_{\mathbf{x}}} p^{-s}(\mathbf{x}') d\mathbf{x}' + \frac{s+\frac{1}{\tau}}{2} p^{-s}(\mathbf{x}) \int_{L_{\mathbf{x}}} p^s(\mathbf{x}') d\mathbf{x}' \right).$$

In order to this hold, the scalar in the parenthesis must vanish,

$$\frac{2}{\tau} vol(L_{\mathbf{x}}) + \frac{1}{\tau} p^s(\mathbf{x}) \int_{U_{\mathbf{x}}} p^{-s}(\mathbf{x}') d\mathbf{x}' - \frac{1}{\tau} p^{-s}(\mathbf{x}) \int_{L_{\mathbf{x}}} p^s(\mathbf{x}') d\mathbf{x}'$$

$$= s p^s(\mathbf{x}) \int_{U_{\mathbf{x}}} p^{-s}(\mathbf{x}') d\mathbf{x}' + s p^{-s}(\mathbf{x}) \int_{L_{\mathbf{x}}} p^s(\mathbf{x}') d\mathbf{x}'.$$

Rearranging the terms give us,

$$\tau = \frac{1}{s} \left( \frac{2 vol(L_{\mathbf{x}}) + 2 p^s(\mathbf{x}) \int_{U_{\mathbf{x}}} p^{-s}(\mathbf{x}') d\mathbf{x}'}{p^s(\mathbf{x}) \int_{U_{\mathbf{x}}} p^{-s}(\mathbf{x}') d\mathbf{x}' + p^{-s}(\mathbf{x}) \int_{L_{\mathbf{x}}} p^s(\mathbf{x}') d\mathbf{x}'} - 1 \right).$$

$\square$

**Theorem 3.1.** *Assume $W \sim Gumbel(0, s)$. A tempering field $\tau(\mathbf{x})$ exists between the belief density $p$ and the MWD $p_w$, and it is given by the formula,*

$$\tau(\mathbf{x}) = s \left( \frac{\int_{\mathcal{X}} \frac{1}{1+r_s(\mathbf{x},\mathbf{x}')} d\mathbf{x}'}{\int_{\mathcal{X}} \frac{r_s(\mathbf{x},\mathbf{x}')}{(1+r_s(\mathbf{x},\mathbf{x}'))^2} d\mathbf{x}'} \right), \tag{B.3}$$

*where $r_s(\mathbf{x}, \mathbf{x}') := p^{\frac{1}{s}}(\mathbf{x}') p^{-\frac{1}{s}}(\mathbf{x})$ is $1/s$-tempered density ratio.*

*Proof.* Under the Bradley–Terry model, $W \sim Gumbel(0, s)$, and the MWD $p_w(\mathbf{x})$ now equals to

$$p_w(\mathbf{x}) = 2\lambda(\mathbf{x}) \int_{\mathcal{X}} F_{\text{Logistic}(0,s)} \left( \log p(\mathbf{x}) - \log p(\mathbf{x}') \right) \lambda(\mathbf{x}') d\mathbf{x}'. \tag{B.4}$$

Following same lines of reasoning as in the constructive proof of Theorem A.1, we fix $\mathbf{x} \in \mathcal{X}$ and aim to find a constant $\tau(\mathbf{x}) = \tau > 0$ solving the tempering field condition. Because $p$ is uniform, the necessary and sufficient condition for the existence of $\tau$ is that it solves the equation,

$$p^{-1-\frac{1}{\tau}}(\mathbf{x}) \int_{\mathcal{X}} \frac{\frac{1}{s} p^{-\frac{1}{s}}(\mathbf{x}) p^{\frac{1}{s}}(\mathbf{x}') - \frac{1}{\tau} \left( 1 + p^{-\frac{1}{s}}(\mathbf{x}) p^{\frac{1}{s}}(\mathbf{x}') \right)}{\left( 1 + p^{-\frac{1}{s}}(\mathbf{x}) p^{\frac{1}{s}}(\mathbf{x}') \right)^2} d\mathbf{x}' = 0.$$

This is equivalent to,

$$\tau = s \left( \frac{\int_{\mathcal{X}} \frac{1}{1+r_s(\mathbf{x},\mathbf{x}')} d\mathbf{x}'}{\int_{\mathcal{X}} \frac{r_s(\mathbf{x},\mathbf{x}')}{(1+r_s(\mathbf{x},\mathbf{x}'))^2} d\mathbf{x}'} \right), \tag{B.5}$$

where for clarity we denote $r_s(\mathbf{x}, \mathbf{x}') := p^{\frac{1}{s}}(\mathbf{x}') p^{-\frac{1}{s}}(\mathbf{x}) = \left( \frac{p(\mathbf{x}')}{p(\mathbf{x})} \right)^{1/s}$, which is $1/s$-tempered density ratio between density values at compared points $\mathbf{x}'$ and $\mathbf{x}$. $\square$

**Proposition 3.2.** *Assume that there exists a tempering field $\tau(\mathbf{x})$ between $p$ and $q$. A tempering parameter $\tau^\star > 0$ defined by,*

$$\tau^\star = \mathbb{E}_{X \sim p}\left(\omega(X)\tau(X)\right), \tag{B.6}$$

*where a stochastic weight $\omega \geq 0$ is given by*

$$\omega(X) = \frac{\|\nabla \log q(X)\|^2}{\mathbb{E}_{X \sim p}\left(\|\nabla \log q(X)\|^2\right)},$$

*minimizes the Fisher divergence between $p$ and $q$,*

$$\tau^\star = \arg\min_{\tau > 0} F(p, q^\tau). \tag{B.7}$$

*Proof.* By the Leibniz formula,

$$
\begin{aligned}
&\frac{\partial}{\partial \tau} \int_{\mathcal{X}} \|\nabla_{\mathbf{x}} \log q^\tau(\mathbf{x}) - \nabla_{\mathbf{x}} \log p(\mathbf{x})\|^2 p(\mathbf{x}) d\mathbf{x} \\
&= \int_{\mathcal{X}} \frac{\partial}{\partial \tau} \|\nabla_{\mathbf{x}} \log q^\tau(\mathbf{x}) - \nabla_{\mathbf{x}} \log p(\mathbf{x})\|^2 p(\mathbf{x}) d\mathbf{x} \\
&= \int_{\mathcal{X}} 2\left(\tau \|\nabla_{\mathbf{x}} \log q(\mathbf{x})\|^2 - \langle \nabla_{\mathbf{x}} \log q(\mathbf{x}), \nabla_{\mathbf{x}} \log p(\mathbf{x})\rangle\right) p(\mathbf{x}) d\mathbf{x}.
\end{aligned}
$$

Since the Fisher score is quadratic in $\tau$, the critical point corresponds to the global minimum. By the assumption, $\langle \nabla_{\mathbf{x}} \log q(\mathbf{x}), \nabla_{\mathbf{x}} \log p(\mathbf{x})\rangle = \tau(\mathbf{x}) \|\nabla_{\mathbf{x}} \log q(\mathbf{x})\|^2$. Hence,

$$\tau^\star = \frac{\mathbb{E}_{X \sim p}\left(\tau(X) \|\nabla \log q(X)\|^2\right)}{\mathbb{E}_{X \sim p}\left(\|\nabla \log q(X)\|^2\right)}.$$

$\square$

**Lemma B.4.** *Let $\tau(\mathbf{x})$ be a tempering field. For any $\tau > 0$ it holds that*

$$F(p, q^\tau) = \int_{\mathcal{X}} |\tau - \tau(\mathbf{x})|^2 \|\nabla_{\mathbf{x}} \log q(\mathbf{x})\|^2 p(\mathbf{x}) d\mathbf{x}.$$

*Proof.* Since $\tau(\mathbf{x})$ is a tempering field,

$$
\begin{aligned}
&\|\nabla_{\mathbf{x}} \log q^\tau(\mathbf{x}) - \nabla_{\mathbf{x}} \log p(\mathbf{x})\| \\
&= \|(\tau - \tau(\mathbf{x}))\nabla_{\mathbf{x}} \log q(\mathbf{x}) + (\tau(\mathbf{x})\nabla_{\mathbf{x}} \log q(\mathbf{x}) - \nabla_{\mathbf{x}} \log p(\mathbf{x}))\| \\
&= \|(\tau - \tau(\mathbf{x}))\nabla_{\mathbf{x}} \log q(\mathbf{x})\|.
\end{aligned}
$$

$\square$

**Proposition B.5.** *Let $\tau^\star$ be the optimal tempering and $\tau(\mathbf{x})$ a tempering field.*

$$F(p, q^{\tau^\star}) = \mathbb{E}\left(\|\nabla \log q(X)\|^2 \tau^2(X)\right) - \frac{\left(\mathbb{E}\left(\tau(X)\|\nabla \log q(X)\|^2\right)\right)^2}{\mathbb{E}\left(\|\nabla \log q(X)\|^2\right)}.$$

*Proof.* By Proposition 3.2 and Lemma B.4,

$$\int_{\mathcal{X}} \left\| \nabla_{\mathbf{x}} \log q^{\tau^\star}(\mathbf{x}) - \nabla_{\mathbf{x}} \log p(\mathbf{x}) \right\|^2 p(\mathbf{x}) d\mathbf{x}$$

$$= \mathbb{E} \left( |\tau^\star - \tau(X)|^2 \left\| \nabla \log q(X) \right\|^2 \right)$$

$$= \mathbb{E} \left( \left( \left( \frac{\mathbb{E} \left( \tau(X') \left\| \nabla \log q(X') \right\|^2 \right)}{\mathbb{E} \left( \left\| \nabla \log q(X') \right\|^2 \right)} - \tau(X) \right)^2 \left\| \nabla \log q(X) \right\|^2 \right) \right)$$

$$= \mathbb{E} \left( \left( \left( \frac{\left\| \nabla \log q(X) \right\| \mathbb{E} \left( \tau(X') \left\| \nabla \log q(X') \right\|^2 \right)}{\mathbb{E} \left( \left\| \nabla \log q(X') \right\|^2 \right)} - \left\| \nabla \log q(X) \right\| \tau(X) \right)^2 \right) \right)$$

$$= \frac{\left( \mathbb{E} \left( \tau(X) \left\| \nabla \log q(X) \right\|^2 \right) \right)^2}{\mathbb{E} \left( \left\| \nabla \log q(X) \right\|^2 \right)} - 2 \frac{\left( \mathbb{E} \left( \tau(X) \left\| \nabla \log q(X) \right\|^2 \right) \right)^2}{\mathbb{E} \left( \left\| \nabla \log q(X) \right\|^2 \right)} + \mathbb{E} \left( \left\| \nabla \log q(X) \right\|^2 \tau^2(X) \right)$$

$$= \mathbb{E} \left( \left\| \nabla \log q(X) \right\|^2 \tau^2(X) \right) - \frac{\left( \mathbb{E} \left( \tau(X) \left\| \nabla \log q(X) \right\|^2 \right) \right)^2}{\mathbb{E} \left( \left\| \nabla \log q(X) \right\|^2 \right)}.$$

$\square$

**Proposition 3.3.** *Let $\tau(\mathbf{x})$ be a tempering field. For any $\tau > 0$ it holds that*

$$F(p, q^\tau) = \mathbb{E}_{X \sim p} \left( |\tau - \tau(X)|^2 \left\| \nabla \log q(X) \right\|^2 \right). \tag{B.8}$$

*Further, when $\tau^\star > 0$ is the optimal tempering*

$$F(p, q^{\tau^\star}) = \mathbb{E}_{X \sim p} \left( \left\| \nabla \log q(X) \right\|^2 \tau^2(X) \right) - \frac{\left( \mathbb{E}_{X \sim p} \left( \tau(X) \left\| \nabla \log q(X) \right\|^2 \right) \right)^2}{\mathbb{E}_{X \sim p} \left( \left\| \nabla \log q(X) \right\|^2 \right)}. \tag{B.9}$$

*Proof.* This is a combined result of Lemma B.4 and Proposition B.5. $\square$

**Corollary B.7.** *Assume that the expert choice model follows the Bradley–Terry model or the exponential RUM. The scores of the belief and the MWD are collinear. That is, there exists a scalar-valued function $\tau(\mathbf{x})$ such that,*

$$\nabla \log p(\mathbf{x}) = \tau(\mathbf{x}) \nabla \log p_w(\mathbf{x}).$$

*Proof.* The result follows directly from Definition 5 and Theorems 3.1 and A.1. $\square$

## C  METHOD

### C.1  TRAINING JOINT AND MARGINALS DISTRIBUTIONS

Recent works discuss in detail how diffusion models can be used to learn between joint and arbitrary conditional distributions, whereas modeling the marginals is not always straightforward (Gloeckler et al., 2024). We adopt a simplified approach to estimate the marginal score function by leveraging a corruption-based marginalization strategy.

To model simultaneously both the joint distribution $p_{\mathbf{x} \succ \mathbf{x}'}(\mathbf{x}, \mathbf{x}')$ and the marginal distribution $p_w(\mathbf{x})$, we introduce a binary conditioning variable joint $\in \{\texttt{true}, \texttt{false}\}$ into the score model. During training, we randomly set joint $= \texttt{false}$ with 50% probability, and in this case, we mask the input $\mathbf{x}'_t$ by replacing it with Gaussian noise $\mathcal{N}(\mathbf{0}, \sigma_t^2 \mathbf{I})$, where $\sigma_t$ is the current noise level. We then compute the denoising score matching loss only over the winner dimensions $\mathbf{x}$ (i.e., the first $d$

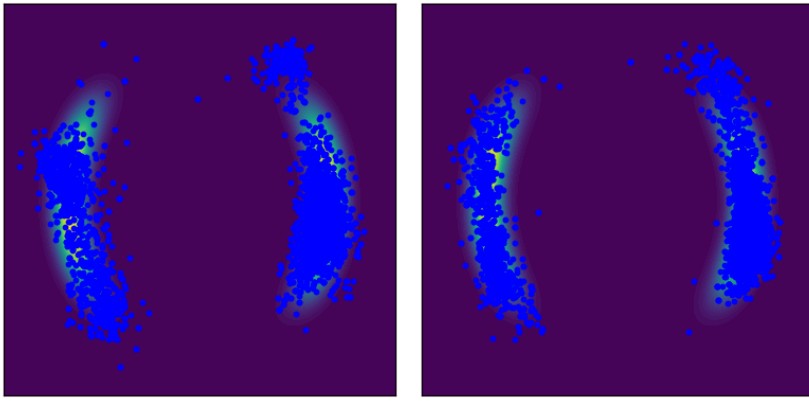

(a) Tempered MWD, w/o joint training      (b) Tempered MWD, w/ joint training

Figure C.1: Replication of Fig. 1 using (a) only winner samples, with the score model trained only for the MWD $p_w(\mathbf{x})$. The quality of the final density estimate, i.e., the 'tempered' MWD, is clearly inferior compared to (b) training on the full joint $p_{\mathbf{x}\succ\mathbf{x}'}(\mathbf{x}, \mathbf{x}')$ using both winners and losers.

Table C.1: Wasserstein distance between the winner marginal samples using $\mathbf{s}_\theta(\mathbf{x}, \mathbf{x}', \sigma, \mathsf{joint} = 0, \mathsf{temp} = 0)$ and $\mathbf{s}_\theta(\mathbf{x}, \mathbf{x}', \sigma, \mathsf{joint} = 1, \mathsf{temp} = 0)$. The last column reports the distance as a percentage relative to the fourth column of Table 1.

| $p(\mathbf{x})$ | wasserstein | relative proportion |
|---|---|---|
| Onemoon2D | 0.085 ($\pm$ 0.009) | 23% |
| Ring2D | 0.077 ($\pm$ 0.012) | 18% |
| Twomoons2D | 0.072 ($\pm$ 0.010) | 18% |
| Mixturegaussians4D | 0.071 ($\pm$ 0.002) | 5% |
| Onegaussian4D | 0.069 ($\pm$ 0.001) | 6% |
| Stargaussian6D | 0.156 ($\pm$ 0.001) | 12% |
| Mixturegaussians10D | 0.366 ($\pm$ 0.001) | 28% |
| Onegaussian16D | 0.671 ($\pm$ 0.001) | 13% |

components of $\nabla \log p_{\mathbf{x}\succ\mathbf{x}'}(\mathbf{x}, \mathbf{x}')$). When $\mathsf{joint} = \mathtt{true}$, we train the model to predict the full joint score over both $\mathbf{x}$ and $\mathbf{x}'$.

At sampling time, to generate samples from the marginal distribution $p_w(\mathbf{x})$ using ALD, we similarly set $\mathsf{joint} = \mathtt{false}$ and replace $\mathbf{x}'_t$ with Gaussian noise $\mathcal{N}(\mathbf{0}, \sigma_t^2 \mathbf{I})$ at each iteration of ALD. Fig. C.1 demonstrates the benefit of training the score model on the full joint data while using the proposed marginalization method to model the marginal score.

**Does the diffusion model learn to marginalize?** To empirically study how well our approach for the diffusion marginalization is able to learn to marginalize, we analyze the difference between the marginal samples from the joint model, and the samples from the marginal model. Specifically, Table C.1 reports the Wasserstein distance between the samples from (i) the true winner marginal of the joint model: $\mathbf{x}$ where $(\mathbf{x}, \mathbf{x}') \sim p_{\mathbf{x}\succ\mathbf{x}'}(\mathbf{x}, \mathbf{x}')$ using the score model $\mathbf{s}_\theta(\mathbf{x}, \mathbf{x}', \sigma, \mathsf{joint} = 1, \mathsf{temp} = 0)$, and (ii) the winner marginal of the joint model: $\mathbf{x}$ where $\mathbf{x} \sim p_w(\mathbf{x})$ using the score model $\mathbf{s}_\theta(\mathbf{x}, \mathbf{x}', \sigma, \mathsf{joint} = 0, \mathsf{temp} = 0)$. To quantify the similarity, we can contrast the Wasserstein distance between these two estimates to the one we have between the target and the score-based estimate (reported in the fourth column in Table 1). This relative difference is typically below 20%, indicating that the marginalization method performs well.

## C.2 Details on modeling the 'tempered' MWD

To learn the MWD $p_w(\mathbf{x})$, having only access to samples from the joint distribution of winners and losers $p_{\mathbf{x} \succ \mathbf{x}'}(\mathbf{x}, \mathbf{x}')$, we propose learning the full joint and the first marginal to capture the preference relationships in the data while enabling sampling from the tempered marginal via score-scaled ALD. To that end, we parametrize the score model $\mathbf{s}_\theta(\mathbf{x}, \mathbf{x}', \sigma, \text{joint}, \text{temp})$ such that $(\mathbf{x}, \mathbf{x}') \mapsto \mathbf{s}_\theta(\mathbf{x}, \mathbf{x}', \sigma_{\min}, \text{joint}=1, \text{temp}=0)$ models the score of the joint distribution of winners and losers, i.e., $\nabla \log p_{\mathbf{x} \succ \mathbf{x}'}(\mathbf{x}, \mathbf{x}')$, and $(\mathbf{x}, \text{temp}) \mapsto \mathbf{s}_\theta(\mathbf{x}, \emptyset, \sigma_{\min}, \text{joint}=0, \text{temp})$ models the score of the MWD and its 'tempered' version, i.e., $\nabla \log p_w(\mathbf{x})$ and $\tau(\mathbf{x})\nabla \log p_w(\mathbf{x})$, respectively. This allows training the score network with both winners and losers, while still enabling sampling from the belief density via the approximation $\mathbf{s}_\theta(\mathbf{x}, \emptyset, \sigma_{\min}, \text{joint}=0, \text{temp}=1) \approx \tau(\mathbf{x})\nabla \log p_w(\mathbf{x}) = \nabla \log p(\mathbf{x})$ by Corollary B.7. Fig. C.1 validates that the joint score model is superior to directly learning the marginal from only the winner samples.

We implement the method by training a score model through the denoising score matching equation 1 on the concatenation of winners and losers, with random masking of losers during training (for more details, see Appendix C.1). Technically speaking, to enable sampling via reverse diffusion *and* ALD, we use a noise distribution $p_{\text{train}}(\sigma)$ during training, defined as a mixture of a Dirac delta on a cosine noise schedule and a $\text{LogNormal}(P_{\text{mean}}, P_{\text{std}}^2)$, with mixture weight $\phi = 0.5$ assigned to the Dirac delta component. We stay as close as possible to the EDM-style diffusion model (Karras et al., 2024a). Specifically, we use the perturbation kernel $p_\sigma(\tilde{\mathbf{x}}|\mathbf{x}) = \mathcal{N}(\tilde{\mathbf{x}}; \mathbf{x}, \sigma^2\mathbf{I})$, which aligns with EDM and defines a forward diffusion process from $\sigma_{\min}$ to $\sigma_{\max}$, where $p_{\sigma_{\min}}(\mathbf{x}) \approx p(\mathbf{x})$ and $p_{\sigma_{\max}}(\mathbf{x}) \approx \mathcal{N}(\mathbf{0}, \sigma_{\max}^2\mathbf{I})$. After training $\mathbf{s}_\theta(\mathbf{x}, \mathbf{x}', \sigma, \text{joint}, 0)$ on perturbed winners and losers, we sample from the belief density using the score-scaled ALD. We can either stop here, or optionally train the tempered marginal score network $\mathbf{s}_{\theta_{\text{MWD}}}(\mathbf{x}, \sigma, \text{temp})$ whose weights can be initialized to that of $\mathbf{s}_\theta$, through denoising score matching using the sampled data. Finally, we use the loss weighting $\ell(\sigma) = \sigma^2$. Algorithms 1–2 summarize the method.

## C.3 Score model

We follow as closely as possible the EDM2 specifications used in the 2D toy experiment in (Karras et al., 2024a). For both the joint score network and the tempered MWD score network, we use an MLP with one input layer and four hidden layers, SiLU activation functions (Hendrycks & Gimpel, 2016) are applied after each hidden layer, and implemented using the magnitude-preserving primitives from EDM2 (Karras et al., 2024b). In the joint score network, the input is a $(2d + 3)$-dimensional vector $(x, x', \sigma, \text{joint}, \text{temp})$, and the output of each hidden layer has $h$ features, where $h \in \{32, 64, 96, 128\}$ depending on the experiment dimensionality. In the MWD score network, the input is a $(d+3)$-dimensional vector $(x, \sigma, 0, \text{temp})$. The binary variables joint and temp are linearly embedded into an $h/4$-dimensional space. Further, a simple residual connection is applied to the embedded variables through all hidden layers. Otherwise, we use the same preconditioning for the score network as described in EDM (Karras et al., 2022).

## C.4 Belief density ratio model

We parametrize the belief density ratio $r_\theta(\mathbf{x}, \mathbf{x}') \approx p(\mathbf{x}')/p(\mathbf{x})$ via parameterizing the unnormalized log density $f_\theta(\mathbf{x}) \approx \log p(\mathbf{x}) + constant$ as an MLP with three hidden layers with SiLu activations, and one output layer, such that $\log r_\theta(\mathbf{x}, \mathbf{x}') = f_\theta(\mathbf{x}') - f_\theta(\mathbf{x})$. The number of hidden units is tied to that of the score model (Section C.3). Regularization of the weights $\theta$ is important for obtaining sensible results. To this end, we apply adaptive $\ell_2$-regularization using the *Adam* optimizer (Kingma & Ba, 2015) with weight decay. In contrast, standard $\ell_2$-regularization, corresponding to *AdamW* (Loshchilov & Hutter, 2019), yielded slightly inferior empirical performance. We set the weight decay to $10^{-3}$, except in the small data $n = 100d$ experiments, where we use a higher value of $3 \times 10^{-3}$.

## C.5 TEMPERING FIELD ESTIMATE

We estimate the integral ratio in Eq. 6 by approximating both the numerator and the denominator using Monte Carlo estimates, using the MWD $p_w(\mathbf{x})$ as importance sampler,

$$\tau(\mathbf{x}) \approx s \left( \frac{\sum \frac{1}{p_w(\mathbf{x}_i)} \frac{1}{1+r_s(\mathbf{x},\mathbf{x}_i)}}{\sum \frac{1}{p_w(\mathbf{x}_i)} \frac{r_s(\mathbf{x},\mathbf{x}_i)}{(1+r_s(\mathbf{x},\mathbf{x}_i))^2}} \right), \tag{C.1}$$

where the sums are over the importance samples $\mathbf{x}_i \sim p_w$. This Monte Carlo ratio estimator is biased, but consistent.

For the $d$-dimensional target, we use $2000d$ importance samples to estimate the integrals in the tempering field. The importance weights are computed using the probability-flow ODE of the MWD diffusion model (see Appendix C.7 for details). The estimated tempering field $\tau(\mathbf{x})$ is clipped such that $1 \leq \tau(\mathbf{x}) \leq Q_\tau(0.99)$, where $Q_\tau(0.99)$ denotes the 99% quantile of the estimated tempering field values (for visualization, we use the 99.9% quantile). The lower bound follows directly from the theory (i.e., from formula 6), while the upper bound is introduced for numerical stability to remove outliers.

## C.6 SCORE-SCALED ALD

Score-scaled ALD uses the scaled score $\tau(\mathbf{x})\nabla \log p_w(\mathbf{x})$, where $\nabla \log p_w(\mathbf{x})$ is replaced by our estimated score. While $\tau(\mathbf{x})$ is not the ALD step size $\epsilon > 0$, it is clear that $\tau(\mathbf{x})$ influences the ALD update in a manner similar to $\epsilon$. To ensure convergence of score-scaled ALD, at each ALD step we use the step size $\epsilon = \frac{\epsilon_{\text{base}}}{\tau(\mathbf{x})} \frac{\sigma^2}{\sigma_{\text{max}}^2}$, where $\epsilon_{\text{base}}$ is the base step size to be specified. The required number of iterations $T$ should be chosen with respect to $\epsilon_{\text{base}}$. In our experiments, we use $L = 50$, $T = 40$, and $\epsilon_{\text{base}} = 0.15$, except in the $2D$-experiments, where we use $\epsilon_{\text{base}} = 7.0$ with $L = 15$ and $T = 40$. In the $2D$-experiments, we keep the original domain scale ($[-3, 3]^d$) and do not rescale it to $[-0.5, 0.5]^d$, unlike in the other experiments.

Regarding the injection of a deterministic ALD noise schedule during denoising score-matching training, we find that the cosine schedule yields better empirical performance, while the noise schedule corresponding to the EDM time-step discretization is also a natural option.

## C.7 DENSITY EVALUATION OF A DIFFUSION MODEL

Chen et al. (2018) showed that for a random variable whose probability density evolves over time, with dynamics $d\mathbf{x} = \tilde{f}(\mathbf{x}_t, t)\,dt$ (where $\tilde{f}$ is Lipschitz continuous in $\mathbf{x}$ and continuous in $t$), the log-density at a point $\mathbf{x}_0$ is given by

$$\log p_0(\mathbf{x}_0) = \log p_T(\mathbf{x}_T) + \int_0^T \nabla \cdot \tilde{f}(\mathbf{x}_t, t)\,dt, \tag{C.2}$$

where $\nabla\cdot$ denotes the divergence operator, which is equal to the trace of the Jacobian. In practice, the divergence is often approximated using the Skilling–Hutchinson trace estimator (Grathwohl et al., 2019),

$$\nabla \cdot \tilde{f}(\mathbf{x}) = \mathbb{E}_{\boldsymbol{\epsilon} \sim \mathcal{N}(\mathbf{0}, \mathbf{I})}[\boldsymbol{\epsilon}^\mathsf{T} J_{\tilde{f}}(\mathbf{x})\boldsymbol{\epsilon}],$$

where the expectation is typically estimated using a finite number of samples.

Now, applying this to EDM-type diffusion model, which is characterized by the probability-flow ODE

$$d\mathbf{x} = -\sigma \nabla \log p_\sigma(\mathbf{x})\,d\sigma, \tag{C.3}$$

we can compute the probability density at a point $\mathbf{x}$ as

$$\log p(\mathbf{x}) \approx \log \mathcal{N}(\mathbf{x}_{\sigma_{\text{max}}}; \mathbf{0}, \sigma_{\text{max}}\mathbf{I}) - \int_{\sigma_{\text{min}}}^{\sigma_{\text{max}}} \sigma \nabla \cdot \nabla \log p_\sigma(\mathbf{x}_\sigma)\,d\sigma, \tag{C.4}$$

where the score $\nabla \log p_\sigma(\mathbf{x})$ is approximated by the score network $\mathbf{s}_\theta(\mathbf{x}, \sigma)$, and the divergence term is estimated using the Skilling–Hutchinson estimator.

The ODE in Eq. C.3 is often stiff at small noise scales, requiring a careful choice of numerical integration method for stable results. To integrate Eq. C.4, we solve the coupled system

$$\frac{d\mathbf{x}}{d\sigma} = -\sigma\,\mathbf{s}_\theta(\mathbf{x}, \sigma), \qquad \frac{d\log p(\mathbf{x})}{d\sigma} = \sigma\,\nabla \cdot \mathbf{s}_\theta(\mathbf{x}, \sigma),$$

where the latter ODE tracks the accumulation of the log-density.

To solve the coupled system, we use the implicit Adams–Bashforth–Moulton black-box ODE solver implemented in the *torchdiffeq* package (Chen, 2018). We compute the divergence exactly using automatic differentiation, although the Skilling–Hutchinson estimator also worked, sometimes even yielding better results. Finally, to further stabilize the density estimates, we clamp the importance weights $1/p(\mathbf{x})$ to the interval defined by their 1st and 90th percentiles.

## D   RUM UNDER THE SPACE REPARAMETERIZATION

This appendix studies the exponential RUM and the Bradley–Terry model under space reparameterization, and verifies that both RUMs are invariant under this transformation. This justifies our approach of transforming possible non-uniform $\lambda(\mathbf{x})$ into a uniform distribution.

The invariance to the space reparameterization holds for Fechnerian RUMs with the winner density,

$$p_w(\mathbf{x}) = 2\lambda(\mathbf{x}) \int_{\mathcal{X}} F\left(\log p(\mathbf{x}) - \log p(\mathbf{x}')\right) \lambda(\mathbf{x}') d\mathbf{x}',$$

where $F$ is a cumulative distribution function of the choice probability. Let a transformation $T$ push the sampling density $\lambda$ into the uniform distribution on the hypercube, that is $T_{\#}\lambda = \mathrm{Unif}([0,1]^d)$. It is enough to show that the winner density in the transformed space has the same form but with uniform sampling density and different belief density. To that end, denote $\mathbf{y} = T(\mathbf{x})$ and note that

$$1 = \lambda(T^{-1}(\mathbf{y}))|\det \nabla T^{-1}(\mathbf{y})|.$$

Hence, $\lambda(\mathbf{x}) = \lambda(T^{-1}(\mathbf{y}))$ in the front of the integral cancels the volume change under the transformation $T$. The integral in the new coordinate system $\mathbf{y}'$ can be computed by the change of variable $\int_{\mathcal{X}} G_{\mathbf{y}} d([T^{-1}]_{\#}\mu) = \int_{[0,1]^d} [G_{\mathbf{y}} \circ T^{-1}] d\mu$ for $G_{\mathbf{y}}(\mathbf{x}') := F\left(\log p(T^{-1}(\mathbf{y})) - \log p(\mathbf{x}')\right)$ and $\mu$ denotes the Lebesgue measure on the hypercube $[0,1]^d$. The winner density in the transformed space reads as

$$p_{w,trans}(\mathbf{y}) = 2 \int_{[0,1]^d} F\left(\log p(T^{-1}(\mathbf{y})) - \log p(T^{-1}(\mathbf{y}'))\right) d\mathbf{y}'.$$

That is, the winner density in the transformed space follows the same RUM than in the original space, but with a uniform sampling distribution and a belief density $p_{trans}(\mathbf{y}) \propto p(T^{-1}(\mathbf{y}))$. Note that the normalization of $p_{trans}(\mathbf{y})$ is irrelevant, as the constant cancels out within the logarithmic utility difference. Consequently, the scale of the RUM noise (parameterized by $F$) remains invariant under the space reparameterization.

# E  EXPERIMENTAL DETAILS

## E.1  TARGET DISTRIBUTIONS

The log unnormalized densities of the target distributions used in the synthetic experiments are provided below.

**Onemoon2D** :
$$-\frac{1}{2}\left(\frac{\|\mathbf{x}\| - 2}{0.2}\right)^2 - \frac{1}{2}\left(\frac{\mathbf{x}_1 + 2}{0.3}\right)^2$$

**Twomoons2D** :
$$-\frac{(\|\mathbf{x}\| - 1)^2}{0.08} - \frac{(|\mathbf{x}_1| - 2)^2}{0.18} + \log\left(1 + e^{-\frac{4\mathbf{x}_1}{0.09}}\right)$$

**Ring2D** :
$$\log\left(\sum_{i=1}^{k}\left(\frac{32}{\pi}e^{-32(\|\mathbf{x}\| - i - 1)^2}\right)\right), \quad \text{where } k = 1$$

**Stargaussian6D** :
$$\log\left(\frac{1}{2}\mathcal{N}(\mathbf{x} \mid \boldsymbol{\mu}, \Sigma_1) + \frac{1}{2}\mathcal{N}(\mathbf{x} \mid \boldsymbol{\mu}, \Sigma_2)\right),$$

$$\sigma^2 = 1,\ \rho = 0.9,\ d = 6,\ \boldsymbol{\mu} = 3\mathbf{1}_d,\ \Sigma_1 = \begin{pmatrix} \sigma^2 & \rho\sigma^2 & \rho\sigma^2 & \cdots & \rho\sigma^2 \\ \rho\sigma^2 & \sigma^2 & \rho\sigma^2 & \cdots & \rho\sigma^2 \\ \rho\sigma^2 & \rho\sigma^2 & \sigma^2 & \cdots & \rho\sigma^2 \\ \vdots & \vdots & \vdots & \ddots & \vdots \\ \rho\sigma^2 & \rho\sigma^2 & \rho\sigma^2 & \cdots & \sigma^2 \end{pmatrix},$$

$$\Sigma_2 = \begin{pmatrix} \sigma^2 & -\rho\sigma^2 & \rho\sigma^2 & \cdots & (-1)^{d-1}\rho\sigma^2 \\ -\rho\sigma^2 & \sigma^2 & -\rho\sigma^2 & \cdots & (-1)^{d-2}\rho\sigma^2 \\ \rho\sigma^2 & -\rho\sigma^2 & \sigma^2 & \cdots & (-1)^{d-3}\rho\sigma^2 \\ \vdots & \vdots & \vdots & \ddots & \vdots \\ (-1)^{d-1}\rho\sigma^2 & (-1)^{d-2}\rho\sigma^2 & (-1)^{d-3}\rho\sigma^2 & \cdots & \sigma^2 \end{pmatrix}$$

**Mixturegaussians**, $d \in \{4, 10\}$ :
$$\log\left(\frac{1}{4}\sum_{i=1}^{4}\exp\left(-\frac{1}{2}(\mathbf{x} - \boldsymbol{\mu}_i)^\top \Sigma_i^{-1}(\mathbf{x} - \boldsymbol{\mu}_i)\right)\right),$$

where $\quad \boldsymbol{\mu}_i = r \cdot \dfrac{\mathbf{v}_i}{\|\mathbf{v}_i\|}, \quad r = 3, \quad \mathbf{v}_1 = \mathbf{1}_d,\ \mathbf{v}_2 = -\mathbf{1}_d,\ \mathbf{v}_3 = \left[(-1)^j\right]_{j=1}^{d},\ \mathbf{v}_4 = -\mathbf{v}_3,$

$\Sigma_i = \mathbf{Q}_i \cdot \mathrm{diag}(\sigma_0^2, \sigma^2, \ldots, \sigma^2) \cdot \mathbf{Q}_i^\top, \quad \sigma_0^2 = 1,\ \sigma^2 = 0.1,\ \mathbf{Q}_i = [\hat{\boldsymbol{\mu}}_i, \ldots] \in \mathbb{R}^{d \times d}$

**Gaussian**, $D \in \{4, 16\}$ :
$$-\frac{1}{2}(\mathbf{x} - \boldsymbol{\mu})^\top \Sigma^{-1}(\mathbf{x} - \boldsymbol{\mu}),\ \boldsymbol{\mu} = 2\begin{pmatrix} (-1)^1 \\ \vdots \\ (-1)^d \end{pmatrix},$$

$$\Sigma = \begin{pmatrix} \frac{d}{10} & \frac{d}{15} & \cdots & \frac{d}{15} \\ \frac{d}{15} & \frac{d}{10} & \cdots & \frac{d}{15} \\ \vdots & \vdots & \ddots & \vdots \\ \frac{d}{15} & \frac{d}{15} & \cdots & \frac{d}{10} \end{pmatrix}$$

## E.2  OTHER EXPERIMENTAL DETAILS

**Hyperparameters and optimization details**. The score models are trained for varying number of iterations ($d = 2 : 8192$, $2 < d < 10 : 12288$, $d \geq 10 : 15360$) with the Adam optimizer (Kingma & Ba, 2015) and a batch size of $\min\{n, 4000\}$ pairwise comparisons, where $n$ is the number of pairwise comparisons in the dataset. For the 2D experiments, we follow (Karras et al., 2024a) and use an adaptive learning rate, specifically a decay schedule of $\alpha_{ref}/\max(\mathrm{iter}, \mathrm{iter}_{ref}, 1)$, with $\alpha_{ref} = 0.005$ and $\mathrm{iter}_{ref} = 1024$ iterations. We use a power-function EMA profile with $\sigma_{ref} = 0.01$. The setup is somewhat sensitive to hyperparameters, and performance can vary depending on their tuning. We expect to achieve better or worse performance in the experiments depending on how well

Table E.1: Robustness to RUM noise family and noise level. The method *score-τ*(**x**) assumes the Bradley–Terry model with noise level $s = 0.7797$, while the data-generating process is varied across three RUMs: Exponential RUM ($W \sim \text{Exp}(s)$), Bradley–Terry ($W \sim \text{Gumbel}(0, s)$), and Thurstone–Mosteller ($W \sim \mathcal{N}(0, s^2)$), each evaluated at noise levels $s \in \{0.1, 0.7797, 1.0, 5.0\}$. Reported values are averages over 10 repetitions of the Wasserstein distance ($\downarrow$).

|  | $s = 0.1$ | $s = 0.7797$ | $s = 1.0$ | $s = 5.0$ |
|---|---|---|---|---|
| **Onemoon2D** |  |  |  |  |
| $W \sim \text{Exp}(s)$ | 0.693 ($\pm$0.078) | 0.266 ($\pm$0.122) | 0.252 ($\pm$0.122) | 0.242 ($\pm$0.117) |
| $W \sim \text{Gumbel}(0, s)$ | 0.358 ($\pm$0.138) | 0.366 ($\pm$0.145) | 0.372 ($\pm$0.149) | 0.599 ($\pm$0.132) |
| $W \sim \text{Normal}(0, s^2)$ | 0.294 ($\pm$0.139) | 0.318 ($\pm$0.124) | 0.321 ($\pm$0.122) | 0.549 ($\pm$0.119) |
| **Twomoons2D** |  |  |  |  |
| $W \sim \text{Exp}(s)$ | 0.577 ($\pm$0.069) | 0.348 ($\pm$0.125) | 0.371 ($\pm$0.127) | 0.383 ($\pm$0.139) |
| $W \sim \text{Gumbel}(0, s)$ | 0.481 ($\pm$0.090) | 0.440 ($\pm$0.089) | 0.446 ($\pm$0.094) | 0.500 ($\pm$0.060) |
| $W \sim \text{Normal}(0, s^2)$ | 0.375 ($\pm$0.144) | 0.345 ($\pm$0.158) | 0.346 ($\pm$0.160) | 0.447 ($\pm$0.080) |
| **Ring2D** |  |  |  |  |
| $W \sim \text{Exp}(s)$ | 0.557 ($\pm$0.071) | 0.423 ($\pm$0.066) | 0.427 ($\pm$0.068) | 0.453 ($\pm$0.089) |
| $W \sim \text{Gumbel}(0, s)$ | 0.462 ($\pm$0.128) | 0.387 ($\pm$0.075) | 0.368 ($\pm$0.074) | 0.449 ($\pm$0.041) |
| $W \sim \text{Normal}(0, s^2)$ | 0.495 ($\pm$0.109) | 0.430 ($\pm$0.086) | 0.420 ($\pm$0.081) | 0.491 ($\pm$0.073) |

the hyperparameters are tuned. The chosen hyperparameters are likely suboptimal, and we expect performance gains, especially in higher-dimensional experiments, if the hyperparameters are well tuned.

**Environment**. All experiments are conducted on a server equipped with nodes containing dual Intel Xeon Cascade Lake processors (20 cores each, 2.1GHz). While exact training times and memory usage were not recorded, the datasets and score network architectures used are relatively lightweight.

**Experiment replications**. Every experiment was replicated with 10 different seeds, ranging from 1 to 10.

**Baseline**. We used the official implementation of (Mikkola et al., 2024) and the provided config files to match the hyperparameter configuration used in their experiments to the closest experiment in our paper. For example, for 2D experiments, we use the config file that was used in their Onemoon2D experiment.

### E.3    ROBUSTNESS TO RUM NOISE FAMILY AND NOISE LEVEL

To study robustness of the method for misspecification in the data-generation process, we rerun Onemoon2D, Twomoons2D, and Ring2D by varying the true data-generation RUM noise family $W$ and the noise level $s$. In all cases, our method assumes the Bradley–Terry model with fixed noise level $s = \sqrt{6/\pi^2}$. Table E.1 reports the results.

The results suggest that lower true noise levels generally lead to higher-quality estimates (note that for $W \sim \text{Exp}(s)$, a smaller $s$ corresponds to a higher noise level). However, using a true noise level that roughly matches the assumed model often yields better or comparable performance than choosing a very small noise level (note that the model is correctly specified only when $W \sim \text{Gumbel}(0, s)$). This observation aligns with the theoretical identifiability of the problem discussed in Section 2.3. Overall, the results appear relatively robust to misspecification of both the noise family and the noise level.

### E.4    RUNTIME BREAKDOWN

The computation times are profiled for the experiments in Section 5. Table E.2 reports the total runtime allocated to three main components: DSM training of the score network (Section 4.1); estimation of the tempering field, including training the density ratio model $r_\theta$ (Section 4.2); and score-scaled ALD sampling (Section 4.3) with the number of samples varying from 25k to 40k. The final column shows the average time required to generate a single sample. In practice, this

also depends on the sampling batch size, which we have not optimized and which is constrained by available memory.

Table E.2: Breakdown of computation times (in seconds). The table reports mean and standard deviation over 10 replicates.

| | DSM training | $\tau(\mathbf{x})$ estimation | sampling total | per sample |
|---|---|---|---|---|
| Onemoon2D | 80 ($\pm$ 16) | 49 ($\pm$ 12) | 690 ($\pm$ 45) | 0.0276 |
| Ring2D | 82 ($\pm$ 15) | 47 ($\pm$ 10) | 692 ($\pm$ 41) | 0.0277 |
| Twomoons2D | 81 ($\pm$ 14) | 48 ($\pm$ 10) | 701 ($\pm$ 42) | 0.0280 |
| Mixturegaussians4D | 262 ($\pm$ 51) | 4560 ($\pm$ 873) | 6958 ($\pm$ 743) | 0.2783 |
| Onegaussian4D | 283 ($\pm$ 46) | 2660 ($\pm$ 279) | 7187 ($\pm$ 404) | 0.2872 |
| Stargaussian6D | 440 ($\pm$ 54) | 3364 ($\pm$ 496) | 12053 ($\pm$ 729) | 0.4018 |
| Mixturegaussians10D | 653 ($\pm$ 47) | 6728 ($\pm$ 1030) | 27314 ($\pm$ 853) | 0.6829 |
| Onegaussian16D | 797 ($\pm$ 24) | 4977 ($\pm$ 223) | 47477 ($\pm$ 727) | 1.1870 |

# F PLOTS

## F.1 PLOTS OF LEARNED BELIEF DENSITIES

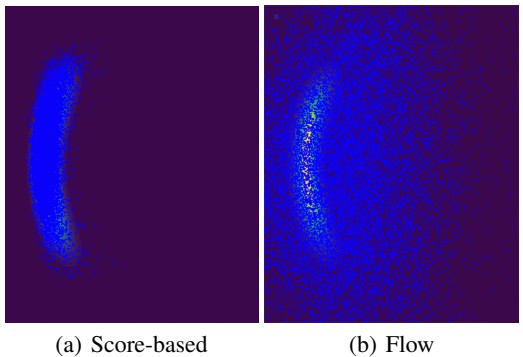

(a) Score-based      (b) Flow

Figure F.1: Onemoon2D experiment. The target distribution is shown as a heatmap, and samples from the learned model are overlaid in blue. (a) Samples from the score-base model. For this particular seed, the estimated tempering field is not too far from the true field, resulting in a good fit. (b) Samples from the flow model.

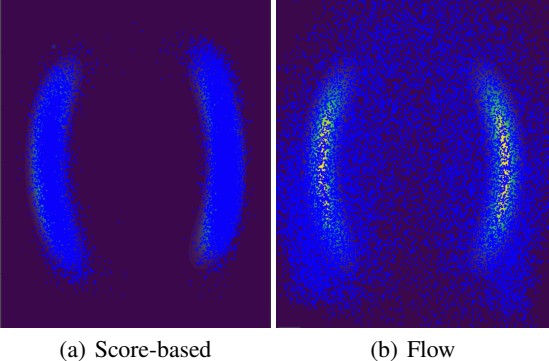

(a) Score-based      (b) Flow

Figure F.2: Twomoons2D experiment. The target distribution is shown as a heatmap, and samples from the learned model are overlaid in blue. (a) Samples from the score-base model. For this particular seed, the estimated tempering field is not too far from the true field, resulting in a good fit. (b) Samples from the flow model.

## F.2 LLM EXPERIMENT

Details of the data generation for the LLM experiment, including the prompts used, are provided in Mikkola et al. (2024, Appendix C.2). We reuse their data and scripts to convert the 5-wise rankings into the pairwise comparisons assumed in our setup: https://github.com/petrus-mikkola/prefflow.

Fig. F.10 completes the partial plot in main text for the LLM experiment. The elicited $2D$ and $1D$ marginals have the same support as the true data distribution marginals, and their shapes are also similar, with the distinction that score-based methods tend to generate Gaussian-like marginals. The only exception is the variable *AveOccup*, whose marginal appears to have an unreasonably long tail.

Fig. F.11 compares our score-based diffusion method and the flow method of learning the LLM prior from pairwise comparisons in the LLM experiment, highlighting that the score-based method results in smoother estimates than the flow method. Table F.1 summarizes the densities in a quantitative manner by reporting the means for all variables.

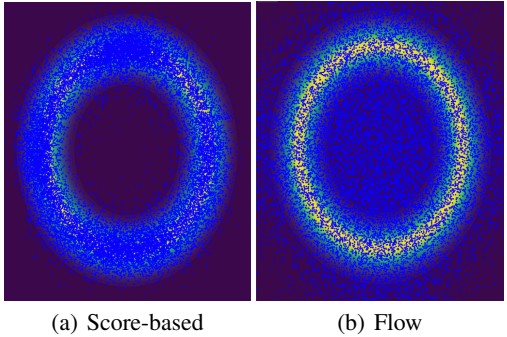

(a) Score-based          (b) Flow

Figure F.3: Ring2D experiment. The target distribution is shown as a heatmap, and samples from the learned model are overlaid in blue. (a) Samples from the score-base model. (b) Samples from the flow model.

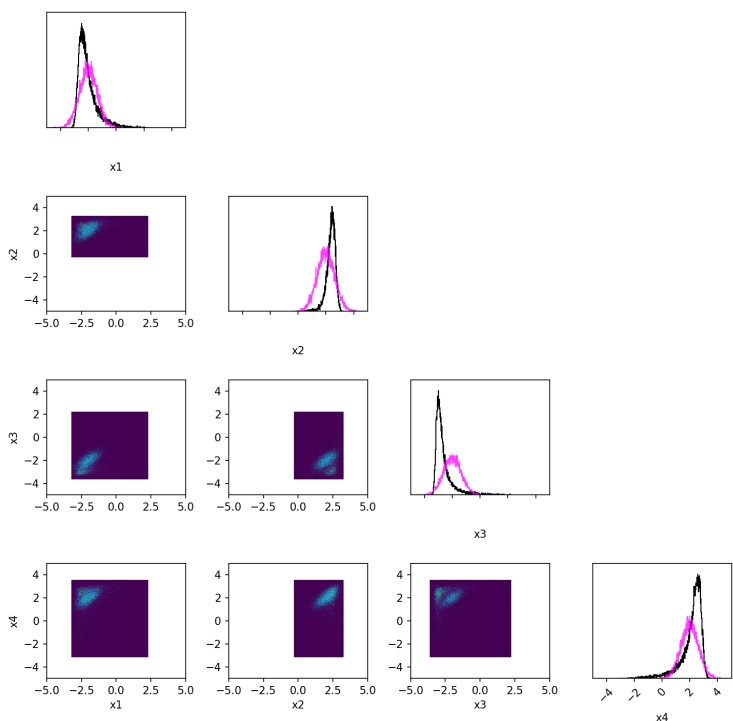

Figure F.4: Gaussian4D experiment. The target distribution is depicted by light blue contour points and its marginal by a pink curve. The learned diffusion model is depicted by greenish blue contour sample points and its marginal by a black curve.

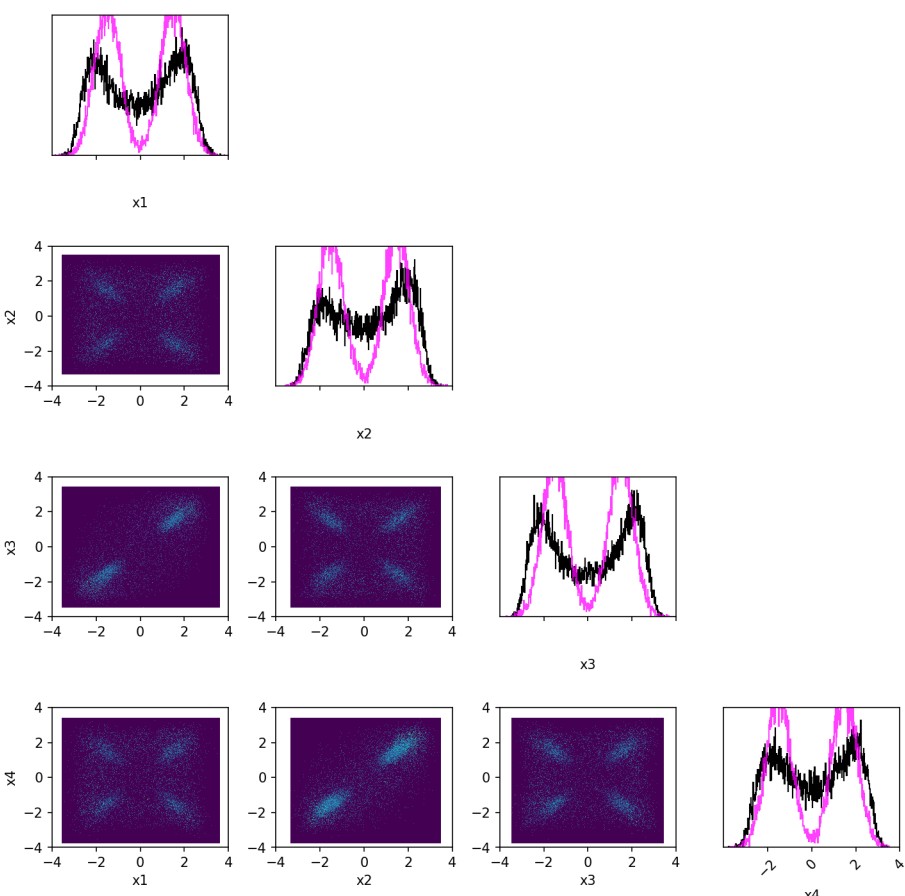

Figure F.5: Mixturegaussians4D experiment. The target distribution is depicted by light blue contour points and its marginal by a pink curve. The learned diffusion model is depicted by greenish blue contour sample points and its marginal by a black curve.

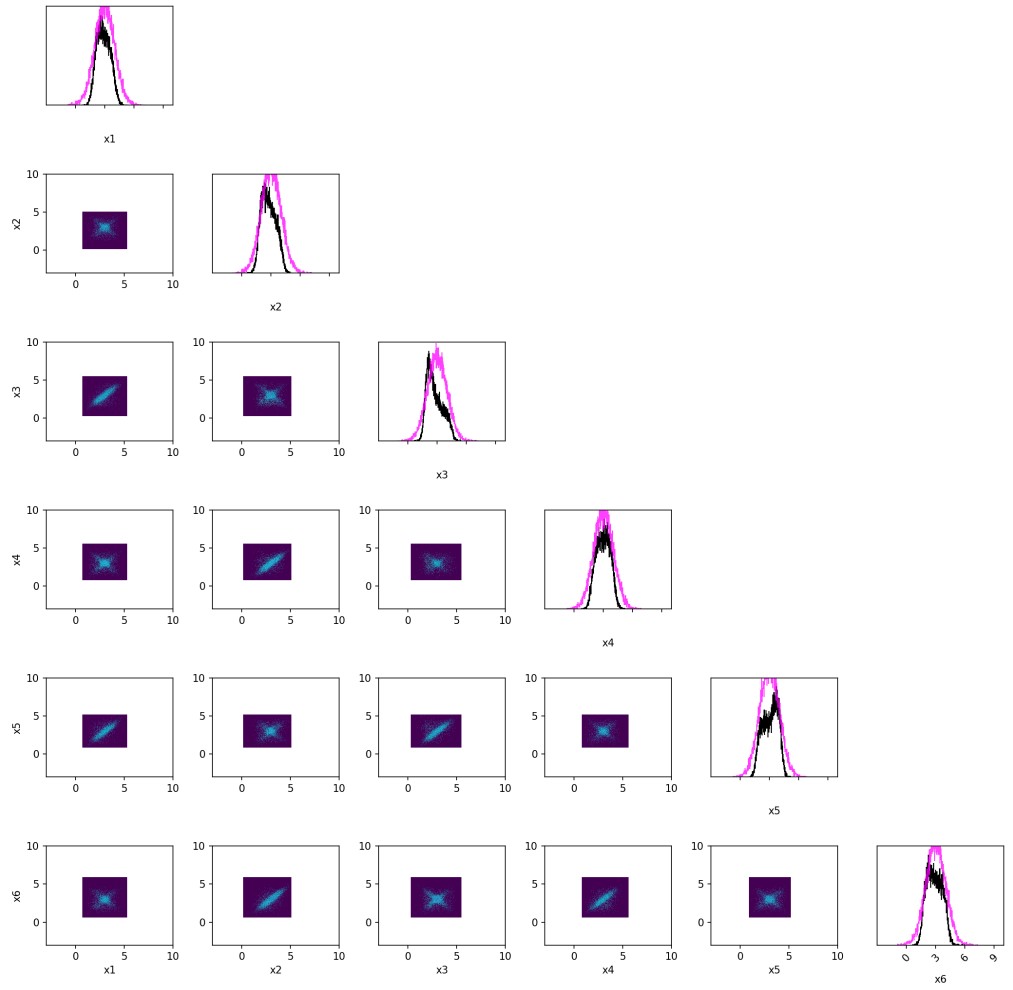

Figure F.6: Stargaussian6D experiment. The target distribution is depicted by light blue contour points and its marginal by a pink curve. The learned diffusion model is depicted by greenish blue contour sample points and its marginal by a black curve.

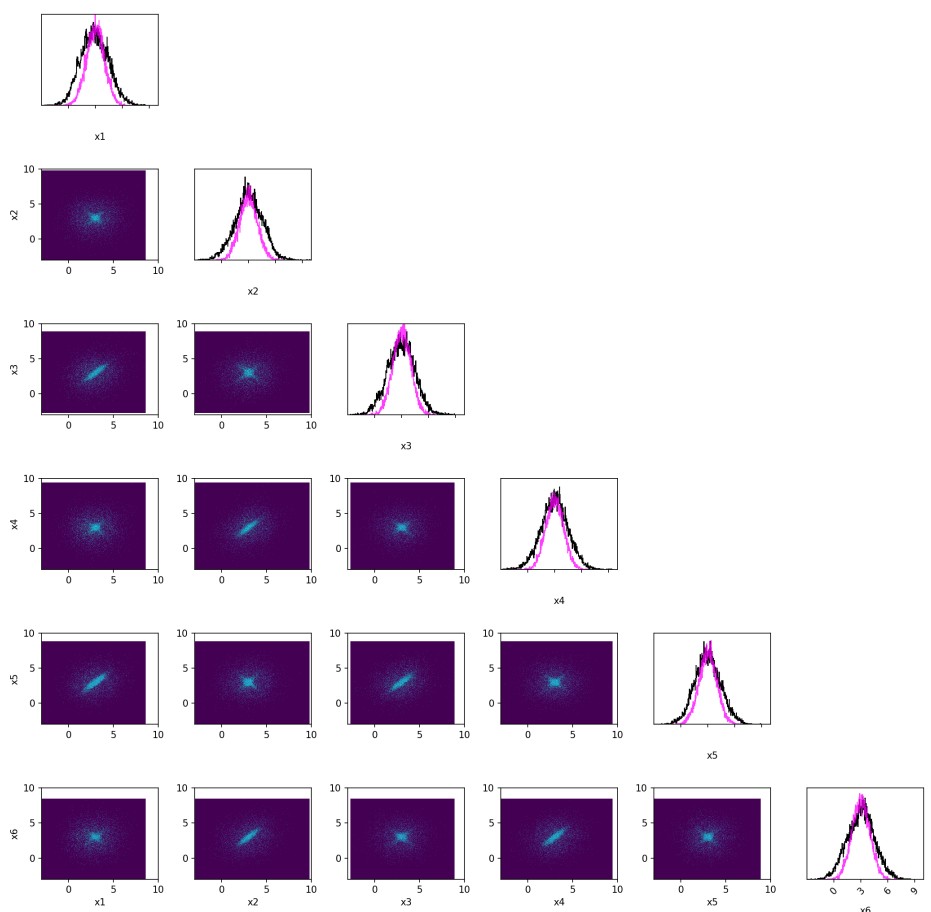

Figure F.7: Stargaussian6D experiment when using the baseline flow-based method. The target distribution is depicted by light blue contour points and its marginal by a pink curve. The learned flow model is depicted by greenish blue contour sample points and its marginal by a black curve.

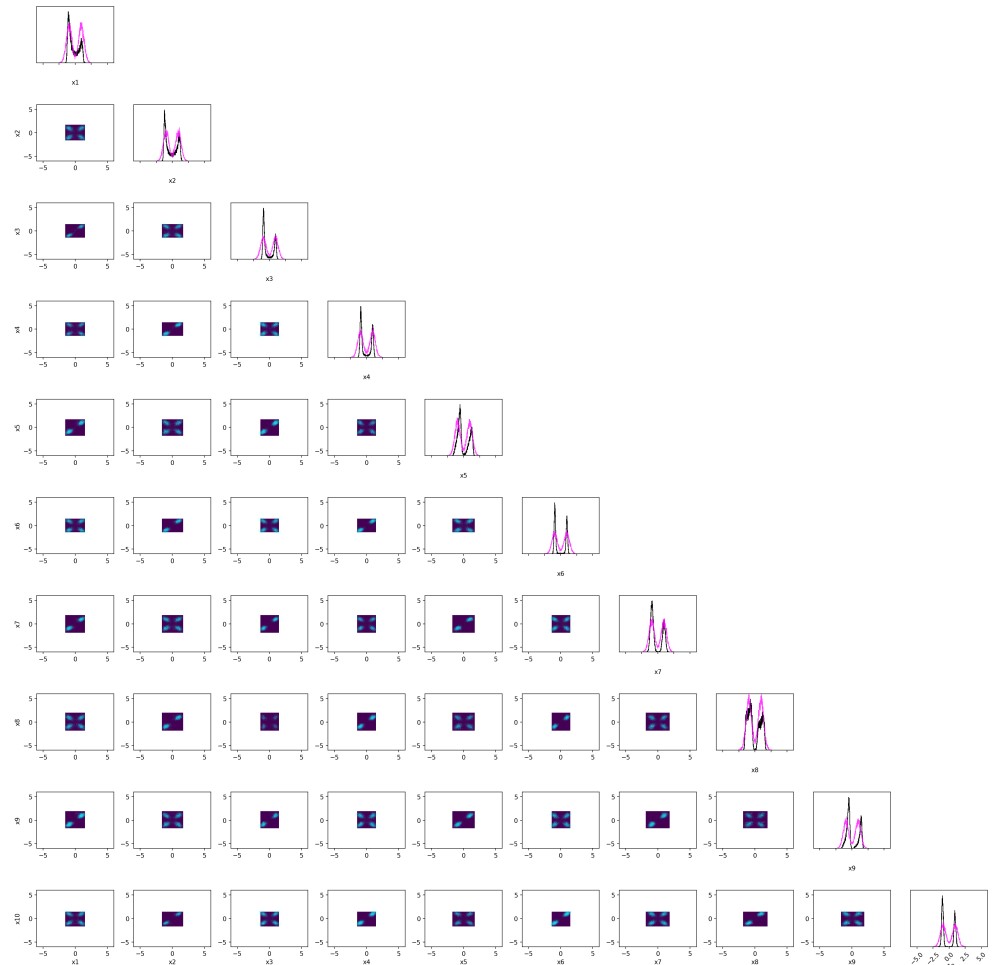

Figure F.8: Mixturegaussians10D experiment. The target distribution is depicted by light blue contour points and its marginal by a pink curve. The learned diffusion model is depicted by greenish blue contour sample points and its marginal by a black curve.

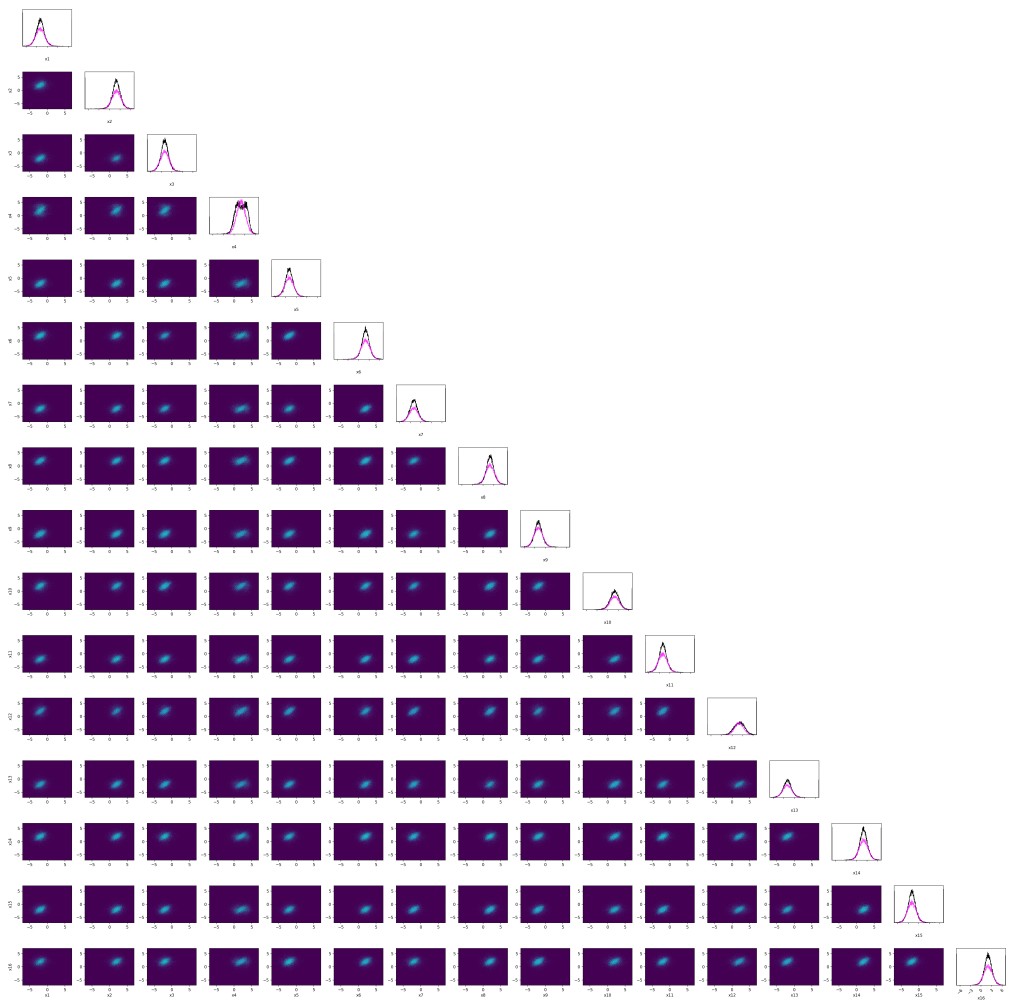

Figure F.9: Gaussian16D experiment. The target distribution is depicted by light blue contour points and its marginal by a pink curve. The learned diffusion model is depicted by greenish blue contour sample points and its marginal by a black curve.

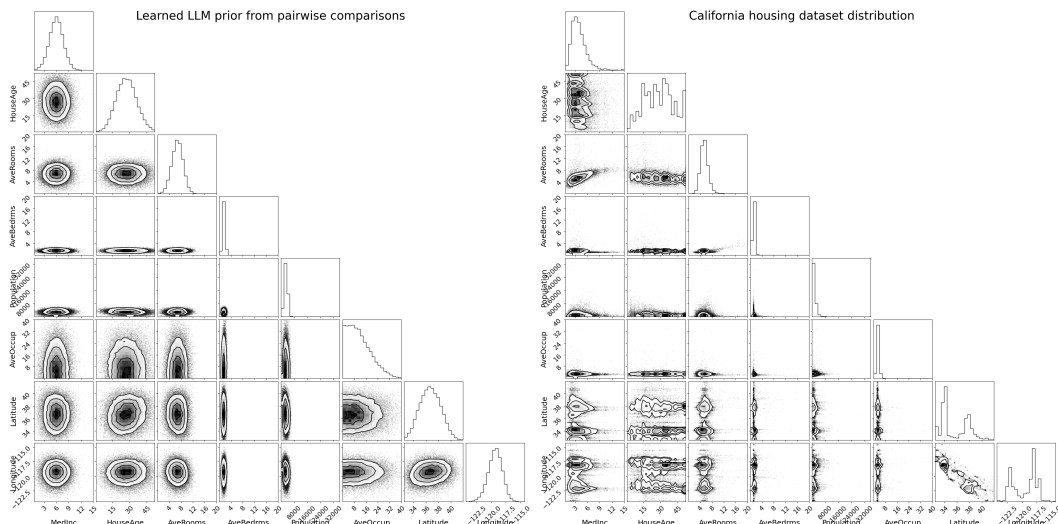

Figure F.10: Full result plot for the LLM expert elicitation experiment, complementing the partial plot presented in Fig. 3.

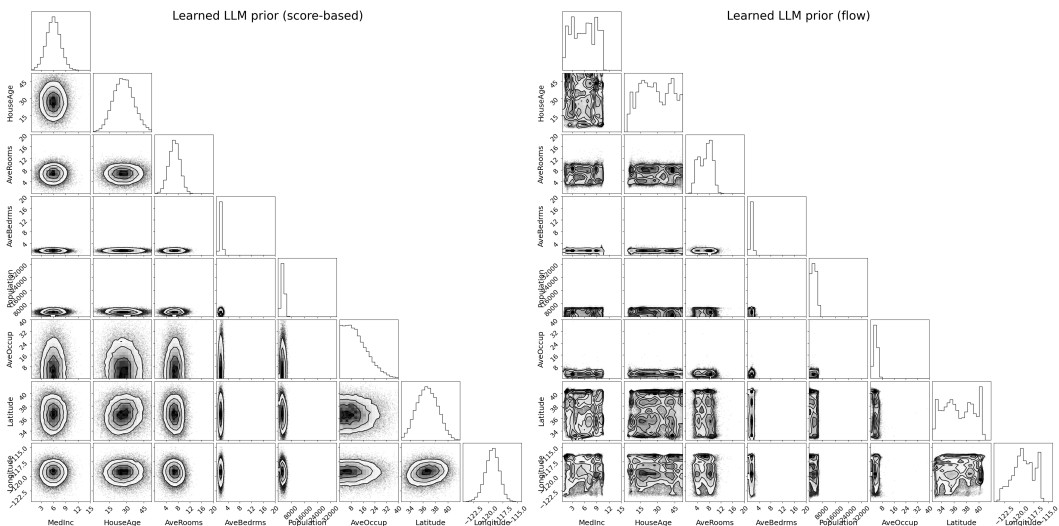

Figure F.11: Comparison of the results when learning the LLM prior from pairwise comparisons using our score-based diffusion method (left) and the flow based method (right).

Table F.1: The means of the variables based on (first row) the distribution of the California housing dataset, (second row) the sample from the score-based diffusion model fitted to the LLM's feedback, and (third row) the sample from the flow model.

|             | MedInc | HouseAge | AveRooms | AveBedrms | Population | AveOccup | Lat   | Long    |
|-------------|--------|----------|----------|-----------|------------|----------|-------|---------|
| True data   | 3.87   | 28.64    | 5.43     | 1.1       | 1425.48    | 3.07     | 35.63 | -119.57 |
| Score-based | 5.89   | 27.56    | 6.66     | 1.53      | 2997.96    | 4.70     | 36.69 | -119.37 |
| Flow        | 5.83   | 28.48    | 6.68     | 1.49      | 2948.17    | 3.36     | 36.73 | -119.30 |

## G   THE USE OF LARGE LANGUAGE MODELS (LLMS)

The data for Experiment 3 "LLM as a proxy for the expert" in Section 5 was obtained by prompting an LLM (Claude 3 Haiku by Anthropic, March 2024). Further, the first version of the Rosenblatt transformation implementation was developed using an LLM. This version was later improved, and the final version was verified to correctly transform points to the hypercube. The inverse transformation also worked in the tested cases. Finally, an LLM was used to check for writing and content errors in both the text and the code.

