# OpenReview forum: "Score-Based Density Estimation from Pairwise Comparisons"
_ICLR.cc/2026/Conference — ICLR 2026 Poster_

### Official Review · Reviewer_zzGP · 2025-10-16

**Soundness:** 2
**Presentation:** 4
**Contribution:** 3
**Rating:** 6
**Confidence:** 3

**Summary:**

This paper provides a method to sample from a distribution given only independent paired samples from an arbitrary distribution and a ranking of pairs of points namely $(x,x',I[U(x)>U(x')])$  where $U$ is some *random* utility function defined as $U(x)=\log p(x)+W$ with $W$ a known noise distribution. It achieves this in two stages: firstly, it learns the score marginal distribution of the sample with highest utility ($p_w(x)$) via denoised score matching, second it learns the transformation $\tau(x)=\frac{\nabla\log p(x)}{\nabla\log p_{w}(x)}$ in order to obtain an estimate for the data distributions score. The paper derives a closed form for $\tau$ as a function of the density ratio when $W$ follows a Gumbel or Exponential distribution as a function of the belief density ratio, $p(x)/p(x')$ for any $x,x'$ pair.

**Strengths:**

*  Well written with clear introductions to each of methods used.

* Tackles an interesting and challenging problem where only pairwise ranks can be used to learn a multivariate distribution.

* Good mix of synthetic illustrative experiments and real world experiments outside of the simplifying assumptions of the paper.

* Provides theoretical justification explanation of  existing method for density estimation with ranked comparisons giving theoretical lower bounds on accuracy of prior work.

**Weaknesses:**

* Estimation of the transformation $\tau$ is non-trivial and relies upon estimating the belief density ratio $r(x,x')=p(x)/p(x')$. Once the belief density ratio has been estimated you already have access to the unnormalised density (for arbitrary fixed $x'$ $p(x)=C\cdot r(x,x'))$ for all $x$) and the score of the distribution ($s(x)=\nabla\log p(x)=\lim_{h\rightarrow 0} \log r(x,x+h)/h$). As such it is not clear that the requirement to learn the MWD and then the tempering field $\tau$ from $r$ is strictly necessary.

* Estimation of $r$ and by consequence $\tau$ is dependent upon the distribution of $W$ being known and from one of two distributional families. This should be made clear in the paper earlier stages of the paper. (It might also be good to highlight that the problem is unidentifiable if $x\succ x'$ is taken as the deterministic density rankings as this is fundamental to the approach and without this the equation on line 79 doesn't seem to make sense as $x\succ x'$ is deterministic.)

* It is unclear how the quality of the method depends upon the accuracy of the assumption on $W$, both in terms of the parameter $s$ and the assumption of the distributional family (namely Gumbel). It would be good to have experiments exploring synthetic data in a more reasonable miss-specified setting.

* The score matching using both winning and paired samples seems slightly under-explored, I appreciate the nice illustration of the improvement from including the paired samples but is there any further justification as opposed to only utilising samples from the distribution you are learning. It seems to me that for this to hold we would reasonably need the neural network to learn that the masking variable represents the marginalisation of the distribution in order for paired samples to meaningfully help learn the MWD.

* (This is more a comment) Algorithms B.1 and B.2 should be in the main body if possible and clarification that the likelihood on line 342 comes from the assumption that $W\~ \text{Gumbel}(0,s)$.

* Small mistake I believe $\mathbb{P}(x\succ x')=F_{W(x')-W(x)}(u(x)-u(x'))$ not $F_{W(x)-W(x')}(u(x)-u(x'))$ (although I could be mistaken.) It's also maybe worth highlighting that this probability is what you mean by the choice distribution conditional on $\mathcal{C}$.


Highlight early on that $x\succ x'$ is random with some fixed noise distribution known up to a parametric family, as this is necessary for the identifiability of the problem.

**Questions:**

* I am unclear of the relationship between 3.1 and the rest of the paper. I appreciate that when $\tau$ is constant you then get the relationship you establish later on ($\nabla \log p=\tau\nabla\log q$) however the later work does not work under the assumption that $\tau$ is constant.

* Can you use the score at the end to produce an explicit density estimate or does it exclusively give a score function and thereby a means of sampling from a distribution.

* Is there any way to use the score estimate to give a new proposal distribution say $\lambda'$ which is closer to $p$?

* Were other choices for estimation of $s_\theta$ explored such as normalising flows?

---

> ### Author Response · Authors · 2025-11-21
> **Response to Reviewer zzGP (1/2)**
>
> We are happy to see that the reviewer both understood the paper well and viewed it positively. We also thank the reviewer for the constructive comments and valuable points, which we address below.
>
> > Estimation of the transformation $\tau$ is non-trivial and relies upon estimating the belief density ratio $r(x,x')=p(x)/p(x')$. Once the belief density ratio has been estimated you already have access to the unnormalised density...
>
> The unnormalized log-density (potential/energy landscape) can be estimated with standard preference models, for example using Gaussian processes [1] or neural networks as often done in the reward modeling setting [2]. We specifically want to find the normalized density, matching the canonical belief elicitation setting. That said, although estimating the belief-density ratio $r(\mathbf{x},\mathbf{x}')=p(\mathbf{x})/p(\mathbf{x}')$ is straightforward via MLE, it is nontrivial in practice, as appropriate regularization is critical for obtaining a stable global ratio estimate, especially in small data settings. For the revision, we extended the discussion of challenges concerning density ratio estimation and its regularization (Sections 4.2 and 6).
>
> > Estimation of $r$ and by consequence $\tau$ is dependent upon the distribution of $W$ being known and from one of two distributional families. This should be made clear in the paper earlier stages of the paper...
>
> We now stress this identifiably fact in the revised paper. Specifically, we extended the discussion: "In this paper, we consider two RUMs, explicitly including the noise level, as it is crucial for identifying $p(\mathbf{x})$. *Exact identification of $p(\mathbf{x})$ requires knowing both the correct noise family and noise level*."
>
> > It is unclear how the quality of the method depends upon the accuracy of the assumption on $W$, both in terms of the parameter $s$ and the assumption of the distributional family (namely Gumbel). It would be good to have experiments exploring synthetic data in a more reasonable miss-specified setting.
>
> We rerun Onemoon2D, Twomoons2D, and Ring2D with varying the true data-generation noise family $W$ and the noise level $s$, while we kept same Bradley-Terry model assumption in our methods. The results suggest that lower true noise levels generally lead to higher-quality estimates. However, using the true noise level that matches the model assumption yields better or comparable performance than simply setting the true noise level to be very small. This observation aligns with the identifiability argument discussed above. Overall, the results appear relatively robust to misspecification of both the noise family and the noise level (for more details, see Appendix E.3., which will be uploaded in a few days).
>
> > ...score matching using both winning and paired samples seems slightly under-explored... is there any further justification as opposed to only utilising samples from the distribution you are learning.... for this to hold we would reasonably need the neural network to learn that the masking variable represents the marginalisation of the distribution in order for paired samples to meaningfully help learn the MWD.
>
> Our intuition is that this due the strong symmetry between winners and losers: the losers carry information about where the *winners are less likely to be*. When the score network also sees winner-loser pairs during training, it can learn a richer shared representation for the winner marginal score field. We now also tested empirically how well the score network learns that the masking variable represents the marginalization of the distribution, following a recommendation from Reviewer aZ9i. Overall, the score network learns to marginalize the joint reasonably well, see the experiment in Appendix (new Table C.1) for details.
>
> > (This is more a comment) Algorithms B.1 and B.2 should be in the main body if possible and clarification that the likelihood on line 342 comes from the assumption that $W\sim\text{Gumbel}(0,s)$.
>
> Since we have an extra page for the revision, we can do this now. We moved the algorithms to the main text and we clarified that the likelihood follows from the assumption $W\sim\text{Gumbel}(0,s)$.
>
> > ...$F_{W(x')-W(x)}(u(x)-u(x'))$ not $F_{W(x)-W(x')}(u(x)-u(x'))$...
>
> You are right. We have fixed that, as well as other typos and unclear phrasing.

---

> > ### Author Response · Authors · 2025-11-21
> > **Response to Reviewer zzGP (2/2)**
> >
> > > I am unclear of the relationship between 3.1 and the rest of the paper. I appreciate that when $\tau$ is constant you then get the relationship...
> >
> > Besides introducing and motivating the concept of a tempering field, our aim was to connect it to prior work that only considered constant tempering. We have now streamlined Section 3.1 to discuss only the main point: since tempering a density with a constant $\tau >0$ is equivalent to multiplying its score by $\tau$, we call a function $\tau(\mathbf{x}) > 0$ a tempering field if the score of resulting "tempered" density can be obtained by multiplying its score by $\tau(\mathbf{x})$.
> >
> > > Can you use the score at the end to produce an explicit density estimate or does it exclusively give a score function and thereby a means of sampling from a distribution.
> >
> > This is a subtle point. Yes, we can evaluate the belief density using the probability-flow ODE. However, it is still somewhat open question whether diffusion models are also strong density estimators [4], meaning that in practice the resulting density estimate $p(\mathbf{x})$ at a point $\mathbf{x}$ is not always reliable. We stress in the revision that, for example, compared to flow-based methods, our approach does not offer an equally efficient or stable way to evaluate the learned belief density.
> >
> > > Is there any way to use the score estimate to give a new proposal distribution say $\lambda'$ which is closer to $p$?
> >
> > Having $\lambda$ that is as similar to $p$ as possible certainly helps reducing the number of preferential comparisons needed for learning the density, and an active learning setup like this would be an intriguing opportunity that we also considered.
> >
> > However, there are quite a few details that would need to be ironed out to deliver a justified method and we had to leave this as future work. It is easy to come up with possible solutions, like using a mixture of the current estimate and an initial broad $\lambda$ as a proposal with some suitable weighting schedule, but harder to find the best way of doing this in practice when we e.g. need to keep track of $\lambda$ for the transformation to uniform. We would also need quite broad experimental validation to ablate the effects of the various choices.
> >
> > > Were other choices for estimation of $s_\theta$ explored such as normalising flows?
> >
> > At the early stages of the project, we experimented with vanilla score-matching variants, but modeling the score across multiple noise scales turned out to be critical for achieving good mode mixing in multimodal targets, consistent with common findings (e.g., [3]).
> >
> > **References**
> >
> > [1] Chu, W., and Ghahramani, Z. (2005). Preference learning with Gaussian processes. ICML2025
> >
> > [2] Ouyang et al. (2022). Training language models to follow instructions with human feedback. NeurIPS2020
> >
> > [3] Song, Y., and Ermon, S. (2019). Generative modeling by estimating gradients of the data distribution. NeurIPS2019
> >
> > [4] Zheng, K., Lu, C., Chen, J., and Zhu, J. (2023). Improved techniques for maximum likelihood estimation for diffusion odes. ICML2023

---

> > > ### Comment · Reviewer_zzGP · 2025-11-25
> > >
> > > Thank you for your comments and clarifications. Please add an additional comment informing me once Appendix E.3 and Table C.1 have been uploaded.

---

### Official Review · Reviewer_aZ9i · 2025-10-29

**Soundness:** 3
**Presentation:** 3
**Contribution:** 3
**Rating:** 6
**Confidence:** 4

**Summary:**

This paper studies density estimation from pairwise comparisons: For example, an expert is shown two i.i.d. draws $\mathbf{x}, \mathbf{x}'$ from a sampling distribution $\lambda(\mathbf{x})$, and the object they prefer is recorded. It is assumed that $\mathbf{x}$ is preferred over $\mathbf{x}'$ with probability $\mathbb{P}(\mathbf{x} \succ \mathbf{x}')$, which is modeled as a random utility model (RUM) with deterministic utility $u(\mathbf{x}) = \log p(\mathbf{x})$. The goal of this work is to infer $p$ from a set of $(\mathbf{x}, \mathbf{x}', \mathbf{x} \succ \mathbf{x}')$ triples. Firstly, for two special cases of RUM (Bradley-Terry & exponential noise RUM), the authors (1) provide theoretical results that relate the marginal winner density (MWD) $p_w(\mathbf{x}) \propto \int \mathbb{P}(\mathbf{x} \succ \mathbf{x}') \lambda(\mathbf{x}) \lambda(\mathbf{x}') \, \mathrm{d}\mathbf{x}'$ to $p(\mathbf{x})$ via a position-dependent *tempering field* $\tau(\mathbf{x})$ by showing that their scores are collinear, i.e., $\nabla \log p(\mathbf{x}) = \tau(\mathbf{x}) \nabla \log p_w(\mathbf{x})$, and (2) provide explicit formulas for $\tau(\mathbf{x})$.
Secondly, the paper builds on these insights by proposing a practical algorithm: The authors train (1) a diffusion model to estimate $\nabla \log p_w$ (MWD model), and (2) a second neural network to estimate the density ratio $r_\theta(\mathbf{x}, \mathbf{x}') \approx p(\mathbf{x}')/p(\mathbf{x})$, which is then used to estimate $\tau(\mathbf{x})$ (with the MWD model as a proposal distribution).
Leveraging the relation to the score of $p$ allows the authors to then draw approximate samples from $p$ using score-scaled annealed Langevin dynamics.
Empirical findings show that this approach outperforms a previous flow-based method qualitatively and quantitatively.

**Strengths:**

+ **Motivation.** The work is well-motivated and tackles an important issue in preference learning. The studied RUM models are relevant in many disciplines.
+ **Theoretical Results.** The theoretical results are sound, well-justified, novel, and important for this line of research. Moreover, they give rise to novel practical algorithms for estimating $p$ using score-based approaches.
+ **Presentation.** The main results are presented in a clear way. It was easy to follow the author's main arguments.

**Weaknesses:**

+ **Estimating $\tau(\mathbf{x})$**
	+ Given a parametric model $r_{\theta}(\mathbf{x}, \mathbf{x}') \approx p(\mathbf{x}')/p(\mathbf{x})$, estimating $\tau(\mathbf{x})$ still involves approximating two (possibly high-dimensional) integrals (Eq. 7) for each $\mathbf{x}$, which is computationally very expensive.
	+ While estimating these integrals with importance-sampled Monte-Carlo (MC) yields unbiased estimates for each of the integrals, the resulting MC estimate for $\tau(\mathbf{x})$ is *not* unbiased, since it involves a fraction of the two integrals. The paper would benefit from a more thorough discussion of this estimation procedure. Moreover, claiming that "the integrals are computed using importance sampling [...]" (L343-L344) is misleading ("computed" should be replaced).

+ **Computational cost for sampling from the estimated $p$**
	+ In score-scaled annealed Langevin dynamics (ALD), using the scaled score $\tau(\mathbf{x}) \nabla \log p_w(\mathbf{x})$ requires estimating $\tau(\mathbf{x})$ in each step of ALD. In Appendix C.6, it is stated that for a $d$-dimensional target, $2000d$ importance samples are used to estimate the integrals in Eq. 7. Hence, a single ALD step is at least $2000d$ times more expensive than regular ALD sampling (where the score is obtained by a single forward pass of the network)---while not counting the computational complexity of calling $r_\theta$ and estimating $\log p_w(\mathbf{x})$ with the probability-flow ODE.
	+ It is clear that this does not scale well to large $d$, or large networks (neither to large diffusion model, nor large $r_\theta$).

+ **Score collinearity when $\sigma > 0$**
	+ The identity $\nabla \log p(\mathbf{x}) = \tau(\mathbf{x}) \nabla \log p_w(\mathbf{x})$ only holds in the *noise-free* case, i.e., $\sigma = 0$. When replacing $p$, $p_w$ with their Gaussian-smoothed version $p*\mathcal N(0,\sigma^2I)$ and $p_w*\mathcal N(0,\sigma^2I)$ for some $\sigma > 0$, this is *not* true anymore. I think this should be stressed more explicitly in the main text.
	+ L353-L354 claims that the "tempering field relation (Eq. 6) is guaranteed to hold in the small-noise limit [...]". This is misleading, as Eq. 6 only holds when $\sigma = 0$ exactly, and is only approximate when $\sigma \approx 0$.
	+ L354: "Possible mismatch at higher noise scales is not an issue [...]". This is only true in the ideal case of $L \to \infty$. In practical settings, where $L$ is moderate, the scores at higher noise levels will have a large effect on ALD.

+ **Hyperparameters**
	+ Many of the empirical results seem sensitive to the particular choice of both training and sampling hyperparameters. In Appendix C.7, it seems that $\epsilon_{\text{base}}$  is chosen very differently for 2D experiments ($\epsilon_{\text{base}} = 7.0$) than in the other experiments ($\epsilon_{\text{base}} = 0.15$). This could hint at the fact that the magnitude of $\tau(\mathbf{x})$ may be miscalibrated.

+ L109: "The network is trained to predict the score of this kernel, which is typically tractable."
	+ This is not true: When minimizing the objective in Eq. (1), the network will learn $s_{\theta}(\mathbf{x}, \sigma) \approx \nabla_{\mathbf{x}} \log p_\sigma(\mathbf{x})$, which is the score of $p_\sigma$, and not of the perturbation kernel $p_\sigma(\tilde{\mathbf{x}} \mid \mathbf{x})$.

+ Very minor suggestions regarding notation:
	+ L163: I suppose $C$ should be a set, so this should read $C = \\{\mathbf{x}, \mathbf{x}' \\}$ instead.
	+ L374-375 reads "2$d$ and 1$d$" marginals, but $d$ refers to the *particular* dimension of the target (L367), so it should rather read e.g. "2D and 1D" (where "D" is used as a shorthand for "dimensional").

+ Typos:
	+ L1022, L1052: "probability ODE", should probably read "probability-flow ODE"
	+ L1049: The equation for the Skilling-Hutchinson trace estimator should read $=$, not $\approx$. The expectation is the exact trace, estimating the expectation with Monte Carlo makes it approximate.

**Questions:**

+ L373 claims that a "linear MLP score network" is used. What exactly is meant with "linear" here? The used MLP is surely a non-linear function of inputs/parameters.
+ Have the authors considered training a model to output $\tau(\mathbf{x})$ directly, e.g. by regressing many (expensive) Monte Carlo estimates? This would amortize the computational complexity during sampling.
+ In low dimensions (e.g., $d = 2$), numerical integration algorithms (e.g. quadrature) have much better convergence rates than Monte Carlo. Have the authors tried such methods to estimate $\tau(\mathbf{x})$ (instead of importance-sampled Monte Carlo)?
+ When training the diffusion model on the full joint (Figure C.1 (b)), is the distribution of the model when `joint = False` actually close to the *true* marginal of the joint model? An experiment would be interesting where (1) you sample $(\mathbf{x}, \mathbf{x}')$ from the model with `joint = True`and discard $\mathbf{x}'$ (which is an *exact* sample from the winner marginal), and (2) you sample $\mathbf{x}$ with `joint = False`, and compare the distributions (e.g. Wasserstein).
+ Why is it that the diffusion model trained on the full joint (both winners and losers) learns "better" winner marginals than the model only trained on winner? I find this surprising.

---

> ### Author Response · Authors · 2025-11-21
> **Response to Reviewer aZ9i (1/2)**
>
> We are pleased that the reviewer both understood the paper well and perceived it positively. We thank the reviewer for the detailed and valuable constructive comments, and provide responses to the specific comments and questions below.
>
> > [...] While estimating these integrals with importance-sampled Monte-Carlo (MC) yields unbiased estimates for each of the integrals, the resulting MC estimate for $\tau(\mathbf{x})$ is not unbiased, since it involves a fraction of the two integrals [...]
>
> This is indeed correct and worth pointing out. We now explicitly mention this in the main text (Section 4.2). The resulting plug-in Monte Carlo estimator of the tempering field is consistent but biased. Similar biased ratio estimators appear in self-normalized importance sampling (Owen, 2013) and in every-visit off-policy value estimation in reinforcement learning (Sutton et al., 1998), where they are preferred due to their favorable variance properties. A first-order bias correction could be considered to reduce the bias from from $O(1/n)$ to $O(1/n^2)$, but we did not go there, since the estimator appeared to work well in practice. We extend the discussion on this.
>
>
> > Computational cost for sampling from the estimated $p$... for a $d$-dimensional target, $2000d$ importance samples are used to estimate the integrals in Eq. 7. Hence, a single ALD step is at least $2000d$ times more expensive than regular ALD sampling (where the score is obtained by a single forward pass of the network)---while not counting the computational complexity of calling $r_\theta$ and estimating $\log p_w(\mathbf{x})$ with the probability-flow ODE.
>
> Note that we sample $\mathbf{X} \sim p_w$ only once and also compute the log-densities $\log p_w(\mathbf{X})$ once using the probability-flow ODE (the Initialization block in Algorithm~2). These values are reused in all forward passes of $\tau(\mathbf{x})$. That is, we only need a single forward pass of $r_{\theta}$ and simple tensor operations to form the MC ratio estimate.
>
> This was sufficient for efficient sampling in our cases, but naturally the approach remains quite a bit slower than the  sampling of the flow-based method that only requires a single pass. We now explicitly mention this in Section 6.
>
> > The identity $\nabla \log p(\mathbf{x}) = \tau(\mathbf{x}) \nabla \log p_w(\mathbf{x})$ only holds in the noise-free case, i.e., $\sigma = 0$. When replacing $p$, $p_w$ with their Gaussian-smoothed version $p*\mathcal N(0,\sigma^2I)$ and $p_w*\mathcal N(0,\sigma^2I)$ for some $\sigma > 0$, this is not true anymore. I think this should be stressed more explicitly in the main text...
>
> We agree. We now stress more clearly that the score collinearity holds exactly only in the noise-free case, and in practice, the mismatch at nonzero noise levels affects ALD sampling. Characterization of the approximation error for non-zero noise scales remains as a future work.
>
> > Many of the empirical results seem sensitive to the particular choice of both training and sampling hyperparameters. In Appendix C.7, it seems that $\epsilon_{\text{base}}$ is chosen very differently for 2D experiments ($\epsilon_{\text{base}} = 7.0$) than in the other experiments ($\epsilon_{\text{base}} = 0.15$). This could hint at the fact that the magnitude of $\tau(\mathbf{x})$ may be miscalibrated.
>
> Good catch -- this particular difference is due to not normalizing the domain in 2D experiments. For the two experiments we used domains of different size ($[-3,3]^D$ when $D=2$ and $[-0.5,0.5]^D)$ when $D>2$) and corrected for the effect this has on the scale of the scores with the base step-size, but we could have written the base step-size directly as a function of the domain scale (that is known). In reality, the method is not particularly sensitive to $\epsilon_{\text{base}}$ that we always kept at a fixed value for all experiments of a given domain size.
>
> That said, we admit that the method is more sensitive to some other hyperparameters such as the regularization parameter of $r_{\theta}$, as discussed in the second last paragraph of Section 6.
>
> > ...The network is trained to predict the score of this kernel... minor suggestions regarding notation...
>
> Thank you pointing these out. We corrected the writing mistakes ('score of this kernel' $\rightarrow$ score of $p_\sigma$, 'linear MLP' $\rightarrow$ MLP, etc.).

---

> > ### Author Response · Authors · 2025-11-21
> > **Response to Reviewer aZ9i (2/2)**
> >
> > > Have the authors considered training a model to output $\tau(\mathbf{x})$ directly, e.g. by regressing many (expensive) Monte Carlo estimates? This would amortize the computational complexity during sampling.
> >
> > This is a good idea and something we considered as well. In effect, we go half-way there, by pre-computing the log-likelihoods for the MC samples but chose not to amortize $\tau(\mathbf{x})$ itself even though it certainly could be done.
> >
> > > In low dimensions (e.g., $d = 2$), numerical integration algorithms (e.g. quadrature) have much better convergence rates than Monte Carlo. Have the authors tried such methods to estimate $\tau(\mathbf{x})$ (instead of importance-sampled Monte Carlo)?
> >
> > Nice idea. We did not try this as we wanted to present a general method that can be used also for larger $d$, but we now mention this as possible improvement for low-$d$ applications.
> >
> > > When training the diffusion model on the full joint (Figure C.1 (b)), is the distribution of the model when joint = False actually close to the true marginal of the joint model? An experiment would be interesting where (1) you sample  from the model with joint = Trueand discard  (which is an exact sample from the winner marginal), and (2) you sample  with joint = False, and compare the distributions (e.g. Wasserstein).
> >
> > This is a good sanity check and we now include a direct comparison in Appendix (new Table C.1). For example, on Twomoon2D, the two distributions are visually near identical. To quantify the similarity, we can contrast the Wasserstein distance between these two estimates to the one we have between the target and the score-based estimate; the difference between the joint and marginal estimates is less than 20\% of that difference.
> >
> > > Why is it that the diffusion model trained on the full joint (both winners and losers) learns "better" winner marginals than the model only trained on winner? I find this surprising.
> >
> > This fact may be initially slightly confusing but then quite intuitive: the losers carry information about where the *winners are less likely to be*. When the score network also sees winner-loser pairs during training, it can learn a richer shared representation for the winner marginal score field, whereas training only on the winners throws away that information.

---

### Official Review · Reviewer_qpz9 · 2025-10-31

**Soundness:** 3
**Presentation:** 2
**Contribution:** 3
**Rating:** 6
**Confidence:** 3

**Summary:**

The paper studies density estimation from pairwise comparisons, linking the unobserved target density to a tempered winner density. By learning the winner’s score through score matching and applying a de-tempering step, the target density can be recovered. The authors prove a collinearity relation between the belief and winner scores, derive a tempering field, and propose an estimator under the Bradley–Terry model. A diffusion-based method using annealed Langevin dynamics demonstrates effective learning of complex belief densities from limited comparisons.

**Strengths:**

1. Estimating the density function from pairwise comparison is novel.
2. The motivation and structure of the work are clear.
3. The theoretical results can show the benefits of the proposed method.

**Weaknesses:**

1. The notation is a little messy. Is there a notation list table to explain the meaning of these items?
2. Maybe it is better to provide an analysis for the efficiency of the proposed method. There should be some experiment results or some discussions on it.
3. I recommend to provide a more clear structure of the algorithm in Section 4, which should be the core contribution of your work.

**Questions:**

Please see Weakness.

---

> ### Author Response · Authors · 2025-11-21
> **Response to Reviewer qpz9**
>
> We thank the reviewer for their constructive comments and positive evaluation of our work. The reviewer's comments are addressed below.
>
> > The notation is a little messy. Is there a notation list table to explain the meaning of these items?
>
> We added the list of notations into Appendix G.
>
> > Maybe it is better to provide an analysis for the efficiency of the proposed method. There should be some experiment results or some discussions on it.
>
> Good point. We did not detail the computational cost of the method as it is not a major element in the kinds of problems we considered, due to low sample counts and dimensionality.
>
> We add the complete breakdown of runtimes into Appendix E.2. For example, in the 2D experiments DSM training took approximately one and a half minutes, and tempering field estimation took less than one minute. The total time is comparable to the flow-based comparison method, whose training required roughly a few minutes. At test time (for sampling and density evaluation) our method is slower than flow-based methods and we revised the discussion to mention this (Section 6). Again, we have not considered the cost any kind of an issue in typical applications.
>
> > I recommend to provide a more clear structure of the algorithm in Section 4, which should be the core contribution of your work.
>
> Thank your for this recommendation. We move Algorithms B.1 and B.2 to the main text. The extra page for the revision also allows us to expand the method section (Section 4.1).

---

### Official Review · Reviewer_NgZe · 2025-11-01

**Soundness:** 2
**Presentation:** 3
**Contribution:** 2
**Rating:** 6
**Confidence:** 3

**Summary:**

In this paper, the authors propose a density estimation method for recovering the target density from pairwise comparison data. The key contribution is showing that the score of the target density is colinear with the score of the winner’s density, allowing it to be estimated via score matching. The proposed method is evaluated on synthetic data generated from a Random Utility Model (RUM) as well as data derived from large language model (LLM) experts, demonstrating promising performance compared with Mikkola et al. (2024).

**Strengths:**

The problem of estimating the density function from expert knowledge is novel in generative model context and underexplored.

The score factorization in (6) is a natural extension of Mikkola et al. (2024)'s proposal where tau(x) here is a constant. The connection with score matching (Fisher divergence) and generative model is very natural as well.

The performance reported in Section 5 is a clear improvement over the normalising flow model used in Mikkola et al. (2024).

The use of LLM to generate expert belief is an efficient way of producing cheap expert comparisons and is a motivating problem setting for the proposed method.

This paper is reasonably well-written (although I think the problem should be better motivated, see below).

**Weaknesses:**

The main weakness is the confusion of the setting considered in this paper. In what application, it is useful to estimate the experts beliefe density? For example, if I have experts opinions {x > x'} where x is a hand-written image and experts are asked copmare images based on how much they look like a digit one. In what application, would I want to estimate p(x)? I guess the user would probably be interested in knowing whether the experts consider a specific image x being 1 or not but I cannot see why it is useful to sample from p(x)?

I could not find much motivation on that in the introduction section, and the experiment section is largely based on toy datasets.
    - In the housing data example, it is unclear what is the "belief about the distribution of features in the California housing dataset". Does it mean, the expert belief on whether the data point is in the dataset or not?

That said, I can see that mathematically, this is an interesting object, as the method seems to estimate the utility function u(x) in RUM model (line 169), which is often unobserved.

In the experiments, the comparisons are made against Mikkola et al. (2024) using a flow model. However, as a benchmark, the authors should also include a diffusion model with \tau(x) treated as a constant, in order to compare the performance gained by incorporating the estimated \tau(x).

**Questions:**

What is the definition of the expert's belief density? why is it useful to generate sample from it?

4.1, I feel the main estimation objective (i.e., the denoising score matching) should be displayed here, showing how the data distribution is constructed from winners and losers and how exactly they are used in the score matching objective.  The current explanations from line 324 - 328 and B1 is not clear enough. For example, what does "train sθ(x,x′,σ,joint) using score-matching" mean?

---

> ### Author Response · Authors · 2025-11-21
> **Response to Reviewer NgZe (1/2)**
>
> We thank the reviewer for their constructive comments, recommendations, and positive evaluation of our work. We address the comments below.
>
> > The main weakness is the confusion of the setting considered in this paper. In what application, it is useful to estimate the experts beliefe density?
>
> Fair point -- we had to resort to a very compact motivation due to lack of space, but can substantially expand and clarify the need with the extra page.
>
> The key setting is that we want to elicit *subjective* belief of an individual (or LLM) over some random variables. The canonical use-case is encoding expert knowledge into statistical models as a prior information, with established literature in statistics dedicated to this problem of *expert knowledge elicitation* [6] or *prior elicitation* [5]. The core assumption is that a domain expert has knowledge about the distribution of plausible model parameters but cannot express it directly as a distribution, and to use that information within a standard statistical modelling workflow we need to elicit it. Here $p(\mathbf{x})$ is a prior over model parameters, and the preference comparisons can be either about the possible parameters directly or about predictive distributions induced by the parameters. We are already working on a follow-up paper that demonstrates the broader methodological relevance of our contribution for the statistics community as a reliable means of eliciting multivariate priors, but for the ICLR audience did not want to elaborate this application too much.
>
> For this audience the most tangible use-cases would perhaps be quantification of LLM's knowledge and probabilistic beliefs, which we now discuss more in the revision. Elicitation tools are increasingly used as principled means of quantifying the LLM knowledge in probabilistic terms, with e.g. [2,7] as recent ICML/NeurIPS papers addressing this. Ability to quantify the LLM beliefs in probabilistic terms helps in measuring calibration and consistency of their internal beliefs, provides ways of combining LLMs with other forms of statistical knowledge, and serves as a drop-in probabilistic model, for example in cognitive models [1] or forecasting models [4]. We show how this can be done based on comparative queries that LLMs can readily answer, without requiring direct prompting of probabilities, samples [7] or moments [2], hence providing a tangible contribution for this active research stream.
>
> > In the housing data example, it is unclear what is the "belief about the distribution of features in the California housing dataset"...
>
> Following on the previous answer, the key point is that we want to quantify the understanding a particular LLM has about the statistical properties of houses in California. We form the *ground truth density* based on the data, but do not use it in any way in the method itself. The California housing problem is just an illustrative example that the method can recover beliefs *and* LLM beliefs are somewhat sensible (aligning with some real data). Given how indirect our approach is, the fact that this works at all is quite interesting (both in terms of our approach and how weirdly poweful LLMs are).
>
> Every human has some understanding of how houses in California are. A Californian real estate would be an expert on this and Californian residents in general would have a better knowledge than e.g. Europeans whose understanding of the houses might come primarily from Hollywood movies, and by eliciting this knowledge we can quantify their understanding. This can be useful, for instance, for verifying whether a politician involved in urban planning has approximately correct understanding of the current situation, or even for measuring the level of expertise of real estate agents.
>
> Now, every LLM has some understanding of this as well, and we offer a method for direct analysis of their representation from a probabilistic perspective. Our method would reveal e.g. if a particular LLM understand the diversity of the housing options correctly, instead of assuming all houses to be like a prototypical one. A comparative study over LLMs would reveal possible biases and the relative accuracies of the models. Our experiments reveal that the particular LLM used here does have a fairly good understanding of the range of house sizes, but is not able to understand the spatial distribution of houses when these locations are communicated solely as longitude and latitude values.

---

> ### Author Response · Authors · 2025-11-21
> **Response to Reviewer NgZe (2/2)**
>
> > However, as a benchmark, the authors should also include a diffusion model with $\tau(x)$ treated as a constant, in order to compare the performance gained by incorporating the estimated $\tau(x)$.
>
> This is a good idea. The revised manuscript includes this additional baseline in Table 1, showing that using the full tempering field $\tau(\mathbf{x})$ is better than resorting to a constant approximation. This also serves as a nice ablation study, isolating the effect of the tempering field and the model for representing the belief from each other. The previous flow-based method considered only constant $\tau$ and we now see our solution is better already without considering the full field.
>
> It is, however, important to note that finding the optimal tempering constant $\tau^\star$ requires estimating the full tempering field $\tau(\mathbf{x})$. This follows from Proposition 3.2 that states $\tau^\star$ is a weighted mean of the tempering field. That is, using a constant approximation does not offer much computational savings.
>
> > 4.1, I feel the main estimation objective (i.e., the denoising score matching) should be displayed here, showing how the data distribution is constructed from winners and losers and how exactly they are used in the score matching objective…
>
> We move Algorithms B.1 and B.2 to the main text, and revise the algorithms and Section 4.1 to make the training procedure easier to follow.
>
>
> **References**
>
> [1] Binz, M., and Schulz, E. (2024) Turning large language models into cognitive models. ICLR2024
>
> [2] Capstick, A., Krishnan, R., and Barnaghi, P. (2025) AutoElicit: Using Large Language Models for Expert Prior Elicitation in Predictive Modelling. ICML2025
>
> [3] Dumoulin, V., Johnson, D. D., Castro, P. S., Larochelle, H., and Dauphin, Y. (2024) A density estimation perspective on learning from pairwise human preferences. TMRL
>
> [4] Halawi, D., Zhang, F., Yueh-Han, C., and Steinhardt, J. (2024). Approaching human-level forecasting with language models. NeurIPS2024
>
> [5] Mikkola, P., Martin, O. A., Chandramouli, S., Hartmann, M., Abril Pla, O., Thomas, O., Pesonen, P., Corander, J., Vehtari, A., Kaski, S., Bürkner, P-C., Klami, A. (2024). Prior knowledge elicitation: The past, present, and future. Bayesian Analysis.
>
> [6] O’Hagan, A. (2019). Expert knowledge elicitation: subjective but scientific. The American Statistician.
>
> [7] Requeima, J., Bronskill, J., Choi, D., Turner, R., and Duvenaud, D. K. (2024). LLM processes: Numerical predictive distributions conditioned on natural language. NeurIPS2024

---

### Author Response · Authors · 2025-11-21

We are pleased the reviewers both understood the paper well and had a positive overall impression. We thank them for their time, effort, and constructive comments, which helped improve both the clarity and soundness of the paper.

The promised experiments are completed, and their results are summarized in the rebuttal. We are now working on integrating the revisions in the full paper, which we will reupload in a few days.

---

> ### Author Response · Authors · 2025-11-25
> **Revision**
>
> We have uploaded the revised manuscript incorporating the reviewers’ comments. The main revisions are summarized below:
>
> + We conducted additional experiments to validate the diffusion marginalization method (Appendix C.1), and an ablation study to demonstrate robustness under RUM noise misspecification (Appendix E.3). We also included a variant of our method using constant tempering to isolate the effect of modeling the full tempering field.
> + We expanded the motivation (Section 1), now including quantification of LLM knowledge in probabilistic terms. We moved the main algorithms to the main text and revised Section 4 to make the training procedure easier to follow. We further clarified the bias of the Monte Carlo estimate, problem identifiability, and the role of regularization in the density ratio model.
> + We stabilized the tempering field estimation (notably importance weight computation, Appendix C.7) and updated the experimental results, which are now improved.
>
> We again thank the reviewers for their constructive comments, nearly all of which have been incorporated directly or indirectly, and which we believe have significantly improved the clarity and soundness of the manuscript.

---

### Author Response · Authors · 2025-12-01
**Summary for the new Area Chair**

For the new Area Chair: Given the exceptional circumstances, we provide a summary of the rebuttal process to support your evaluation.

The only response to our rebuttal came from **Reviewer zzGP** who thanked our comments and asked us to inform when the revision with the two additional experiments was ready, which we uploaded within half a day. The experiments address (i) robustness of our method to model misspecification and (ii) a sanity check of the proposed marginalization method. The results show that the method is reasonably robust to model misspecification (Appendix Table E.3) and that the marginalization method works as intended (Appendix Table C.1).

Our summary of **Reviewers NgZe,qpz9,aZ9i**, who had not yet responded, and our corresponding responses is as follows:

+ *Unclear motivation* (Reviewer NgZe): We expanded the motivation (Section 1), elaborating especially the task of quantifying LLM knowledge in probabilistic terms, with citations to several recent works studying alternative methods for this. This extends the original motivation, which focused on expert knowledge elicitation and general connections to learning from human feedback.
+ *Unclear notation and a request to add an analysis of efficiency* (Reviewer qpz9): We added a list of notation (at the beginning of the Appendix) and provided a runtime breakdown (Appendix E.4).
+ *Clarify bias in the tempering field estimate, a computation cost concern, and stress that theory applies only to original (noiseless) scores* (Reviewer aZ9i): We now stress that the estimate is biased but consistent (Section 4.2), and that score-collinearity holds only in the noiseless case (Section 4.3), although we show empirically that the approximation works well in ALD sampling. Concerning computational cost, we clarified that importance sampling is required only once, so there is no additional cost beyond the forward pass of the neural network $r_{\theta}$ at sampling time.

All revisions are marked in blue in the uploaded manuscript.

---

### Meta-Review · Area_Chair_xoTA · 2026-01-07

**Summary:**

This paper studied the density estimation problem from pairwise comparisons. All reviewers provide positive assessments. Review comments acknowledge that this paper is well-written and achieves a clear improvement compared to the state-of-the-art results. Based on these assessments, I agree with reviewers and recommend acceptance.

**Reviewer Concerns:**

Most reviewer concerns have heen addressed by the rebuttal, such as  new experiments on more diverse datasets and new analysis for fidelity/diversity tradeoffs

**Reviewer Scores:**

There is no clear evidence to suggest that any reviewer would have changed their score after a full discussion.

---

### Decision · Program_Chairs · 2026-01-26

Accept (Poster)